# Response of soil viral communities to land use changes

Hu Liao[1,2,4], Hu Li[1,4], Chen-Song Duan [1,2], Xin-Yuan Zhou[1,2], Qiu-Ping Luo[1,2], Xin-Li An[1], Yong-Guan Zhu[1,3] & Jian-Qiang Su [1,2] ✉

Soil viruses remain understudied when compared to virus found in aquatic ecosystems. Here, we investigate the ecological patterns of soil viral communities across various land use types encompassing forest, agricultural, and urban soil in Xiamen, China. We recovered 59,626 viral operational taxonomic units (vOTUs) via size-fractioned viromic approach with additional mitomycin C treatment to induce virus release from bacterial fraction. Our results show that viral communities are significantly different amongst the land use types considered. A microdiversity analysis indicates that selection act on soil vOTUs, resulting in disparities between land use associated viral communities. Soil pH is one of the major determinants of viral community structure, associated with changes of in-silico predicted host compositions of soil vOTUs. Habitat disturbance and variation of soil moisture potentially contribute to the dynamics of putative lysogenic vOTUs. These findings provide mechanistic understandings of the ecology and evolution of soil viral communities in changing environments.

Soils are intrinsically diverse partially due to their wide compositional spectrum and spatial heterogeneity in terms of physicochemical properties[1], supporting a high diversity of interacting microbes that are pivotal to ecosystem services including global C and N biogeochemical cycles[2]. Viruses are extremely abundant and diverse biological entities on earth, playing vital roles in affecting soil microbiota and functions[3,4] via regulating microbial community dynamics[5], reprogramming host metabolism during infection[6,7], and serving as vectors of horizontal gene transfer[3]. Viral metagenomics has greatly expanded the viral ecology of aquatic ecosystems[8,9] and mammalian guts[10,11], knowledge about soil viruses lags behind. There were approximately a hundred of reported soil viromes derived from Arctic permafrost[12], agricultural soils[13,14], mangrove sediment[4], peat[15], and Antarctic soils[16], providing insights into the niche partitioning of vOTUs[15], their response to climate change and environmental stress[12], and phage mediated horizontal gene transfer[3]. However, none of these viromes was from forest, urban park and road verge soil, the macrodiversity and microdiverisity of viral communities in soils with varied land use types have not been fully explored.

Human activities increasingly change the earth's landscape. It was estimated that nearly a third of global land area were affected by land use change in sixty years[17]. Rapid urbanization often involves changes in land use types, including deforestation, arable farming, and conversion of natural habitat to urban land uses. The habitat disturbance and land use change are profoundly affecting soil microbiome due to the change of vegetation, loss of biodiversity, and input of man-made chemicals. Many studies have documented significant impacts of land use changes on soil microbial communities, functional traits and soil functions[18–20]. Since bacteria are the host of phages that dominate soil DNA virome[21], it is expected that the distribution of soil viral communities could considerably differ across various land use types.

Viruses are classified as lytic or lysogenic according to their strategies of life[11]. There were conflicting studies about the proportion of lysogenic and lytic viruses across different habitats[22]. Some studies

[1]Key Laboratory of Urban Environment and Health, Ningbo Observation and Research Station, Institute of Urban Environment, Chinese Academy of Sciences, 361021 Xiamen, China. [2]University of Chinese Academy of Sciences, 100049 Beijing, China. [3]State Key Lab of Urban and Regional Ecology, Research Center for Eco-environmental Sciences, Chinese Academy of Sciences, 100085 Beijing, China. [4]These authors contributed equally: Hu Liao, Hu Li. ✉e-mail: jqsu@iue.ac.cn

showed that coinfection would lead to frequent lysogenization, especially in highly productive environments due to higher microbial density and higher phage adsorption rate[23,24]. While other studies observed that lysogenic phages were at lower abundance in response to the eutrophic conditions[25]. These results suggest that host associated and abiotic factors strongly influence the switch between lytic and lysogenic strategies[26]. The changes of environmental factors will also exert selection pressures on soil viral operational taxonomic unit (vOTUs), resulting in variant microdiversity. Recent investigations into the viral microdiversity have provided insights into the evolutionary trends of vOTUs in response to geographic distance and depth in marine environments[9,27], while microdiversity of soil vOTUs has not been explored.

Anthropogenic uses of land have converted and fragmented ecosystems, decreased biodiversity[28,29], disturbed soil biogeochemical cycles[30], and caused soil pollution, leading to significant changes in soil biotic and abiotic environmental factors. However, the ecology and evolution of soil viral communities, and interactions with their host in response to such stressors induced by land use changes remained largely unknown. To address these knowledge gaps, we conducted an in-depth characterization of soil viral composition and spatial distribution across five land use types including forest, paddy field, vegetable field, urban park, and road verge through viromic analysis. Host-linked interactions, lysogenicity, and microdiversity were further investigated to illustrate the ecological and evolutionary adaptation of soil virome to land use changes.

## Results

### Characteristics of LVD dataset and assembled vOTUs

The land use virome dataset LVD was derived from 2.6 billion paired clean reads of sequences across 50 viromes of 25 samples with five types of land uses (Supplementary Data 2). A total of 6,442,065 contigs (>1500 bp) were yielded, of which 764,466 (11.8%) contigs were identified as putative viral genomes through VIBRANT. Subsequently, putative false positive viral genomes were removed (see Methods section), and 27,951 and 48,936 bona fide viral genomes were retained from the 25 intracellular VLPs (iVLPs) and 25 extracellular VLPs (eVLPs) viromes, respectively. These genomes were clustered into 25,941 and 45,152 vOTUs for iVLPs and eVLPs viromes, respectively, in which the iVLPs and eVLPs viromes shared 11,467 (19.2%) vOTUs. Subsequently, they were merged and dereplicated, resulting in 59,626 vOTUs (Supplementary Data 3) for the following analysis. A total of 8112 (13.6%) vOTUs genomes were classified as complete, in which the median length of all and circular vOTUs were 25,183 bp and 45,511 bp, respectively (Supplementary Fig. 4).

To explore the taxonomic affiliation of vOTUs in family and genus-level, a gene-sharing network consist of 59,626 vOTUs genomes from this study and 3502 reference phage genomes (from NCBI Viral RefSeq version 201) revealed 6009 VCs comprising of 37,224 vOTUs, of which 34,417 vOTUs were from LVD, besides 2794 singletons (2653 from LVD dataset), 16,056 outliers (15,833 from LVD) and 8492 overlaps (8061 from LVD) were detected (Supplementary Data 4). Of these, only 157 VCs contained genomes from both the RefSeq and LVD dataset (1864 viral genomes) (Supplementary Data 4). Most of VCs (1837, 30.4%) included only two members.

At the family level, most of vOTUs were classified into *Siphoviridae* (712 by vConTACT2 and 29,671 (50.9%) by Demovir, tailed dsDNA), *Podoviridae* (610 by vConTACT2 and 9923 (17.6 %) by Demovir, tailed dsDNA), *Myoviridae* (485 by vConTACT2 and 5445 (9.9%) by Demovir, tailed dsDNA), *Tectiviridae* (50 by vConTACT2 and 10 (0.10%) by Demovir, non-tailed dsDNA) (Fig. 1). Besides, the Eukaryotic viruses *Herpesviridae* (159 by Demovir, 0.26%, dsDNA), *Phycodnaviridae* (120 (0.20%) by Demovir, dsDNA); the Virophage Family *Lavidaviridae* (15 (0.03%) by Demovir) were detected as well, but a majority of vOTUs were unclassified in genus-level.

### Viral community structures differ across land use types

Bray–Curtis dissimilarity of viral communities (median 0.9951) showed strong heterogeneity of viral communities among different sites (Fig. 2a). While, the Bray–Curtis dissimilarity (median: 0.5109) between paired viral communities of iVLPs and eVLPs from each site have a significant lower heterogeneity than inter-sites (Wilcox.test, $p < 0.001$; Fig. 2a). Furtherly, the Bray–Curtis distance between paired intracellular and extracellular viral communities did not present a significant difference among all types of land uses (Supplementary Fig. 5). Similarly, viral communities of paired iVLPs and eVLPs were grouped together for each site and were well separated from the other sites (anosim.test, $R = 0.04$, $p > 0.05$; Fig. 2b). Therefore, the paired iVLPs and eVLPs viromes from each site were merged for subsequently viral community analysis.

PCoA based on Bray–Curtis dissimilarity of viral communities indicated the viral communities of the 25 samples from five types of land use were clustered into three zones (Fig. 2b). We designated these three emergent land use zones as the agricultural zone (AG) including paddy and vegetable plot, urban green spatial zone (UG) including park and road verge, and forest zones (FO) respectively (anoism.test, $R = 0.72$, $p = 1e-4$). PERMANOVA indicated that the best predictor of β-diversity of viral community composition was the land use zones compare with other environmental parameters, which explained 10.4% of the variance (adjust $p < 0.001$) (Supplementary Data 5). The proportion of shared vOTUs within land use zones was significantly greater than those between land use zones (Wilcox.test, $p < 0.0001$; Fig. 3a). The heatmap indicated that the viral communities were clustered according to the land use zones (Fig. 3b) except sample T5 rather

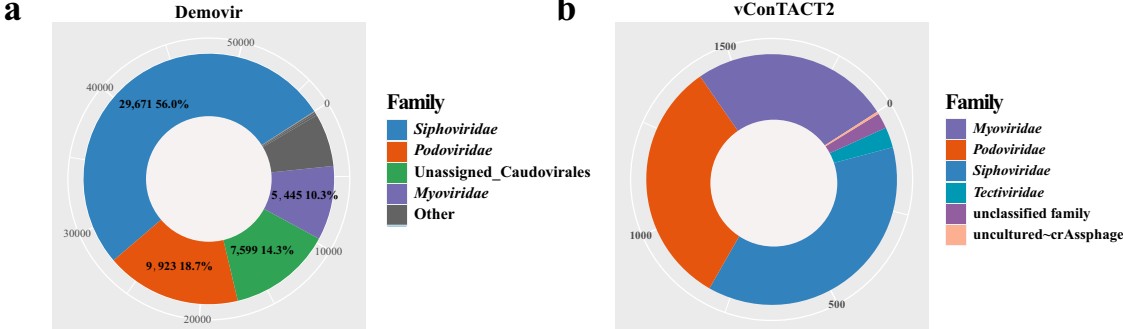

**Fig. 1 | The taxonomic assignment of LVD.** Pie charts showing the affiliation of 56,870 vOTUs at family level assigned by script Demovir (**a**). and the affiliation of 1864 vOTUs at family level assigned by package vConTACT2 (**b**). Source data are provided in the Source Data file.

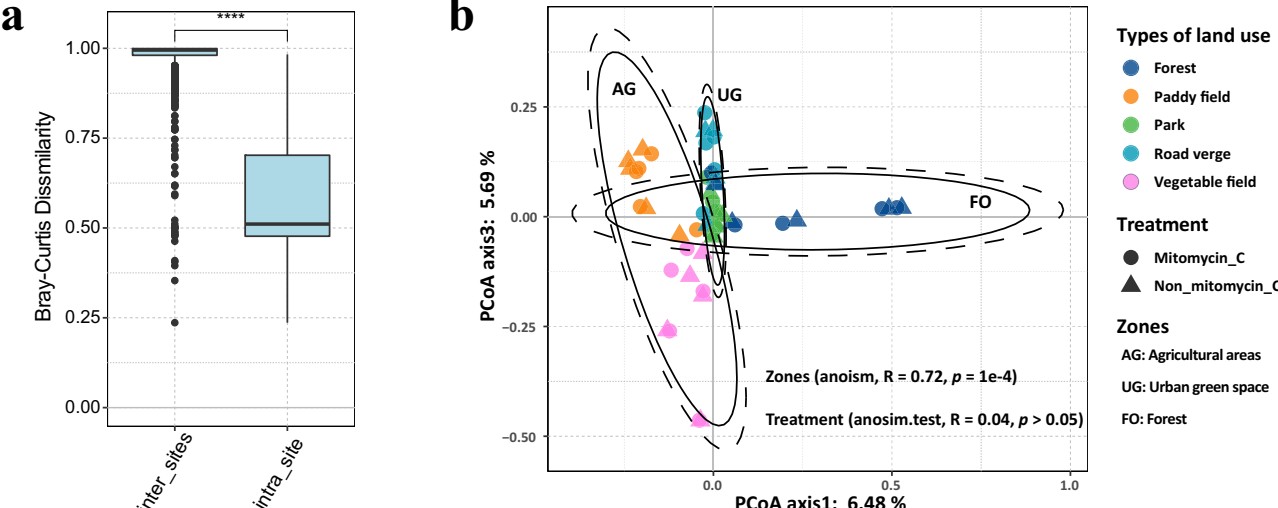

**Fig. 2 | The macrodiversity of soil viral communities. a** Boxplot showing Bray–Curtis dissimilarity of viral communities of intra-sites (between the corresponding community of iVLPs and eVLPs, $n = 25$) and inter-sites (between different sample sites, $n = 300$). The minima, maxima, center, bounds of box and whiskers in boxplots from bottom to top represented percentile 0, 10, 25, 50, 75, 90, and 100, respectively, the difference between different zones was tested using the two-sided Wilcox.test, ****$p < 0.0001$. The exact $p$ value is 1.6e-15. **b** Principal coordinates analysis (PCoA) of viral community structures, as derived from reads mapping to 59,626 vOTUs and Bray–Curtis dissimilarities; each point is one sample, triangle indicated extracellular (Non_mitomycin_C) and circles indicated intracellular (mitomycin_C treatment) viral community. The analysis of similarity (ANOSIM) statistics considered viral community composition grouped by habitat and treatment. The land use zone AG represents agricultural areas including paddy and vegetable field; UG represents urban green space including park and road verge; FO represents forest. Ellipses in the PCoA plot are drawn around the centroids of each zone at 95% (inner) and 97.5% (outer) confidence intervals. The statistical test used was two-tailed. Source data are provided in the Source Data file.

than geographic distance (Supplementary Fig. 6a). A stronger distance-decay pattern was observed within zone FO compared with the other two zones (Supplementary Fig. 6b). Furthermore, at the VCs level, the multi-zonal VCs were dominant and accounted for 24.7% to 45.2% relative abundance (Fig. 3c), suggesting that the strong vOTUs boundary among different zones of land use was less pronounced at VCs level, such as the zone FO and AG did not share any vOTUs, but they shared 34% VCs (Fig. 3d). We also observed that UG and AG shared the most VCs (47%), followed by UG and FO (42%), and FO and AG (32%; Fig. 3d).

The diversity of viral communities of three land use zones were comparable as indicated by Shannon (6.1–7.0) and Simpson (0.9878–0.9977) indexes (Supplementary Fig. 7). The compositions of viral communities were similar, dominated by tailed dsDNA bacteriophages assigned to the family *Siphoviridae*, followed by *Podoviridae*, *Myoviridae*, *Herpesviridae*, and *Tectiviridae* (Supplementary Fig. 8). Mantel test revealed that viral community structures at population-level significantly correlated with 22 environmental parameters, of which the pH played the most important role in the change of viral community structures (Fig. 3e).

### The relative abundances of putative lysogenic phages vary among different land uses

The putative temperate vOTUs were identified based on the presence of integrases or whether the vOTU was identified as integrated into a bacterial contig through VIBRANT. The relative abundance of putative lysogenic phage presented a significant difference among the three land use zones (Fig. 4a), in which the relative abundance of putative lysogenic vOTUs of the AG (from 4.3% to 17.4%) was significantly lower than those in the UG (from 7.5% to 31.0%) and FO (from 24.5% to 47.8%) (Fig. 4a). Additionally, the difference of putative lysogenic phage between paired iVLPs and eVLPs viromes presented an obvious variety across the three zones, of which the iVLPs viromes of FO harbored higher putative lysogenic phages (Fig. 4a). The alpha-diversity of putative lysogenic phage showed a significant difference among the

three zones (Fig. 4b). Mitomycin C treatment can facilitate the recovery of putative lysogenic vOTUs (Wilcox test: $p < 0.001$; Fig. 4c). The relative abundances of putative lysogenic vOTUs were negative correlated with soil moisture (F-statistic: $R^2 = 0.27$, $p < 0.05$; Fig. 4d), whereas a positive correlation with DOC was observed (F-statistic: $R^2 = 0.35$, $p < 0.05$; Fig. 4d). Furthermore, we detected 617 prophages, of which 525 prophages were used as representative genomes of vOTUs, only 18 non-prophage vOTUs contains prophage. Consistent with relative abundance of putative lysogenic phage, we also found the mitomycin C can enhance the recovery of prophage from viromes (Fig. 4e).

### Host-linked viral community compositions

The variations of prokaryotic community composition of different land uses were characterized based on taxonomic profiles derived from corresponding metagenomes. The results showed that the dominant class was *Actinobacteria*, *Bacilli*, *Alphaproteobacteria*, *Betaproteobacteria*, *Gammaproteobacteria*, and *Deltaproteobacteria* (Fig. 5a). There was a significant difference among the five land use types (anoism.test $R = 0.44$, $p < 0.001$; Supplementary Fig. 9a), whereas the alpha-diversity were similar (Supplementary Fig. 9b).

The relative abundances of vOTUs were summed up according to their predicted hosts at class level (Fig. 5b and Supplementary Data 6). The results indicated obvious variations in community compositions according to zones (Fig. 5b) compared with overall soil bacterial compositions (Fig. 5a). Overall, the dominant predicted host were *Actinobacteria*, *Alphaproteobacteria*, *Gammaproteobacteria*, *Betaproteobacteria*, and *Acidobacteriia* (Fig. 5b). The zone UG occupied the highest relative abundance (from 17.5% to 52.8%, Wilcox.test: $p < 0.01$) of *Actinobacteria*-linked phages; the AG occupied the highest relative abundance (from 6.8% to 26.9%, Wilcox.test: $p < 0.05$) of *Betaproteobacteria*-linked phages (Fig. 5b). We explored the host of 8,910 vOTUs with significant differences among the three zones (Supplementary Data 7). Phages infected *Gammaproteobacteria* were enriched in zone FO, and the phages infected *Actinobacteria* and *Alphaproteobacteria*

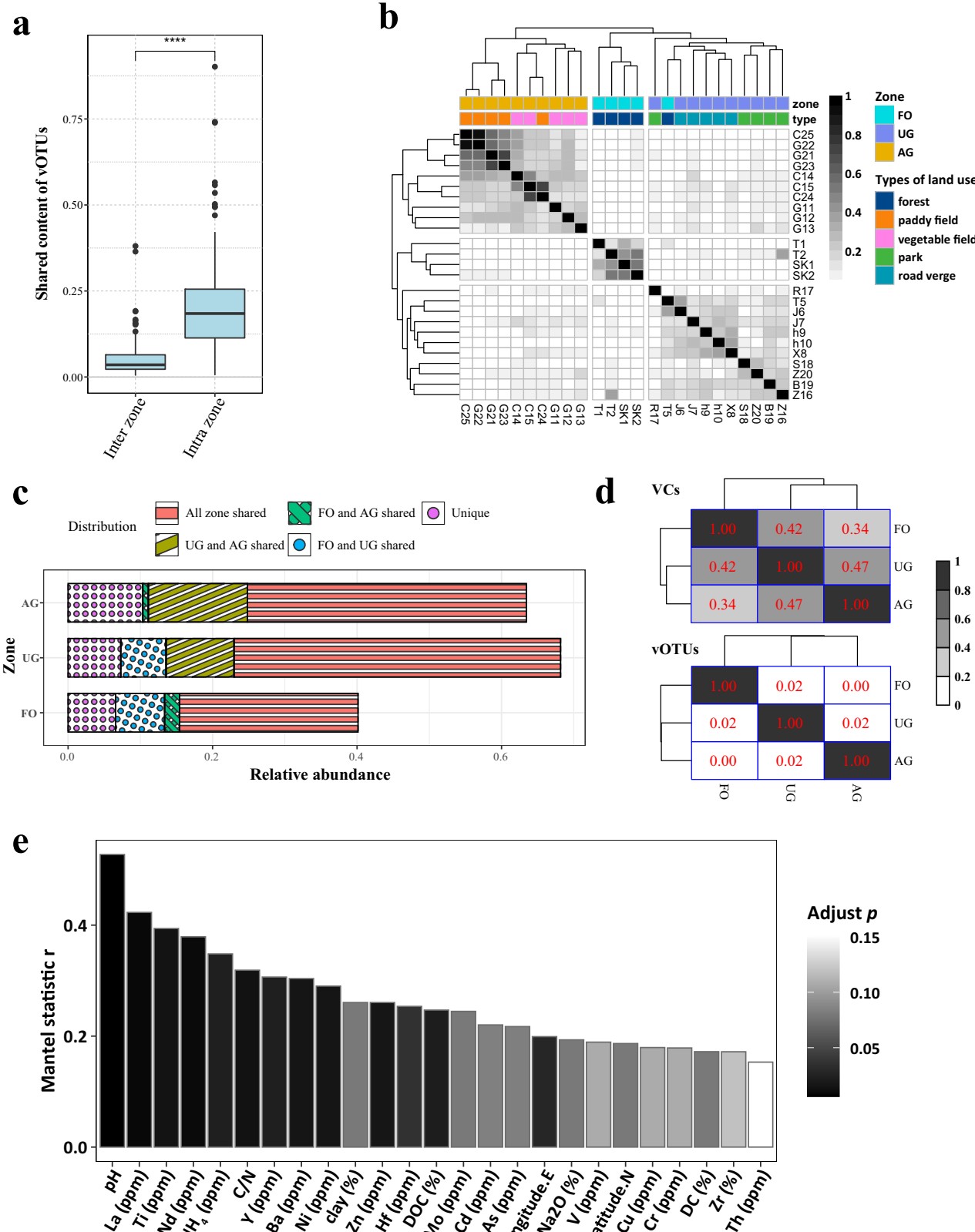

were enriched in zone UG, whereas the zone AG enriched phages that infected *Acidobacteriia* and *Betaproteobacteria* (Supplementary Fig. 10).

Mantel tests revealed that the compositions of bacterial communities were significantly related to the environmental factors DC/DN (Dissolved carbon/Dissolved nitrogen), C/N, DN, $NO_3$, Se, and

longitude (Fig. 5c). The host-linked viral communities were significantly related to bacterial communities (mantel statistic $r = 0.44$, $p < 0.001$), and they were significantly sensitive to the changes of geographic distance (longitude), soil pH, and concentration of metal element Ba, Ni, As, Mo, La, Nd, and Zn (Fig. 5c). The network based on the Spearman correlation coefficient indicated that pH, longitude, P,

**Fig. 3 | Dynamic of viral operational taxonomic unit (vOTUs) and clusters (VCs).**
**a** The shared vOTUs of inter-zones (*n* = 285) and intra-zones (*n* = 30). The minima, maxima, center, bounds of box and whiskers in boxplots from bottom to top represented percentile 0, 10, 25, 50, 75, 90, and 100, respectively, the difference between different zones was tested using the two-sided Wilcox.test, ****$p$ < 0.0001. The exact $p$ value < 2.2e-16. **b** Heatmap showed the shared viral content between different samples. The FO represent forest zone, the UG represent urban green space zone, and AG represent agricultural zone. **c** The relative abundances of VCs according to their distribution. The "all zone shared" as the VCs presented in all zones; "FO and AG shared" as the VCs only presented in zone FO and AG, "UG and AG shared" as the VCs only presented in zone UG and AG; "FO and UG shared" as the VCs only presented in zone FO and UG; "Unique" as the VCs exclusively presented in each zone. **d** The proportion of shared VCs (above) and vOTUs (below) between different zones. **e** Mantel test showing the correlation between viral communities and geochemical parameters. The $p$ values were adjusted using FDR. The statistical test used was two-tailed. Source data are provided in the Source Data file.

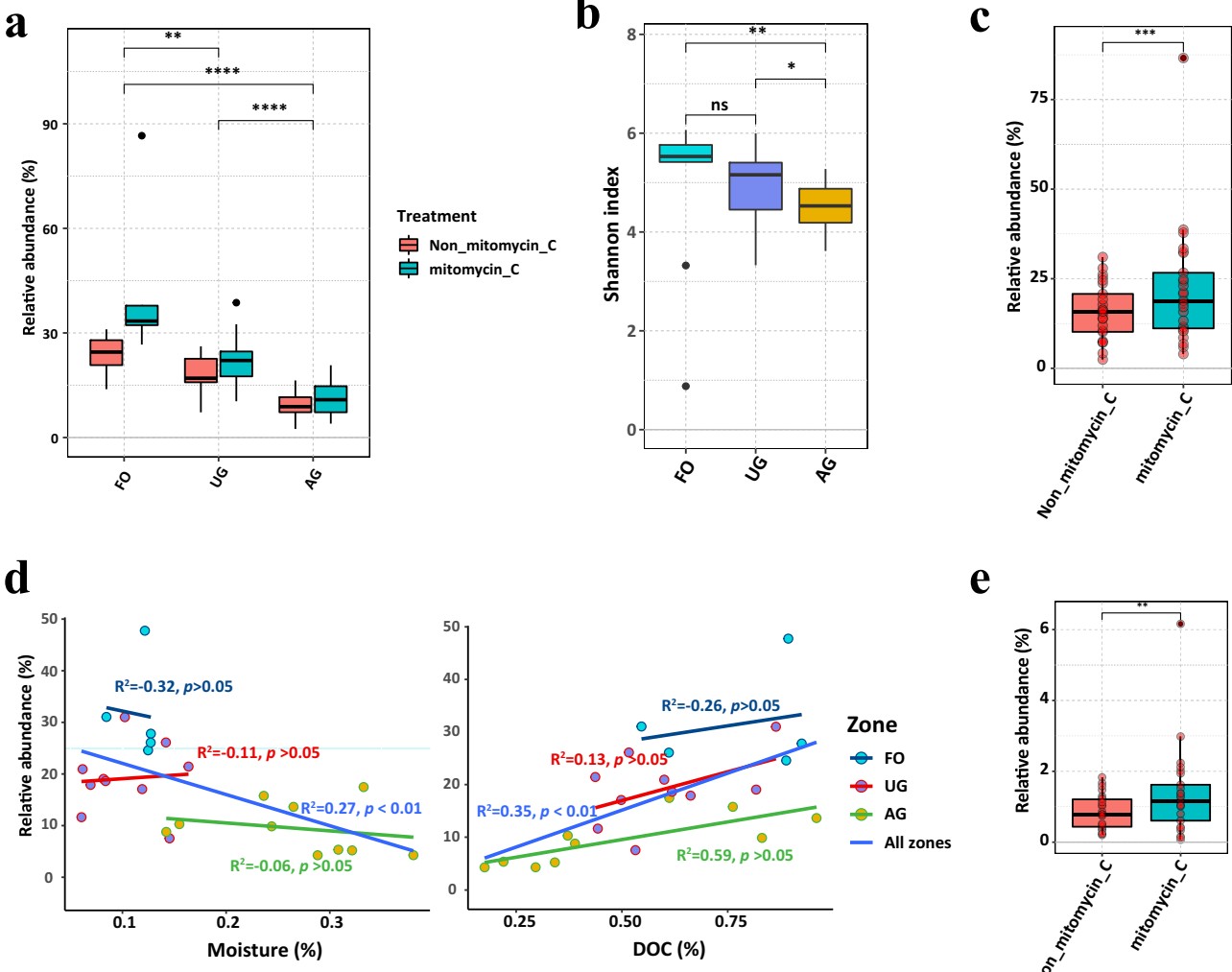

**Fig. 4 | The lifestyle of viruses in different land uses.** The relative abundance (**a**), and the alpha diversity (**b**) of putative lysogenic phage in zones FO (*n* = 5), AG (*n* = 10) and UG (*n* = 10), and (**c**) the relative abundance of putative lysogenic phages in eVLPs (Non_mitomycin_c, *n* = 25) and iVLPs (mitomycin_c, *n* = 25) viromes. The minima, maxima, center, bounds of box and whiskers in boxplots from bottom to top represented percentile 0, 10, 25, 50, 75, 90, and 100, respectively, the difference between different zones was tested using the Wilcox.test, ****, ***, **, * and ns represents $p$ < 0.0001, $p$ < 0.001, $p$ < 0.01, $p$ < 0.05, and $p$ > 0.05 respectively. **d** Relationship between the relative abundance of putative lysogenic phages and moisture and DOC. The dark blue lines represent fitting curve of linear model for all sites. Inset values display the $R^2$ and adjusted $p$ value of F-statistic. The statistical test used was two-tailed. **e** The relative abundance of proviruses in eVLPs (Non_mitomycin_c, *n* = 25) and iVLPs (mitomycin_c, *n* = 25) viromes. The minima, maxima, center, bounds of box and whiskers in boxplots from bottom to top represented percentile 0, 10, 25, 50, 75, 90, and 100, respectively, the difference between different treatments was tested using the Wilcox.test, ** represents $p$ < 0.01. The exact $p$ value is 0.0056. The statistical test used was two-tailed. Source data are provided in the Source Data file.

Zn, and moisture have a higher connection degree compared to the other environmental factors (Supplementary Fig. 11). pH strongly correlated to the phages infecting *Actinobacteriota*, *Acidobacteriia*, *Gammaproteobacteria*, but not to the abundance of these taxa in soil metagenome (Supplementary Fig. 12). The relative abundances of *Actinobacteria*-linked phages were positively correlated with the elevated pH, whereas the relative abundances of *Acidobacteria* and *Gammaproteobacteria*-linked phages were negatively correlated with pH (Supplementary Fig. 12).

## Shared vOTUs and microdiversity
The shared vOTUs based on their distribution within and between land use zones were classified as local, regional, and multi-zonal vOTUs (Fig. 6a). In total, 44,534 vOTUs (74.6%) were classified as "local", and

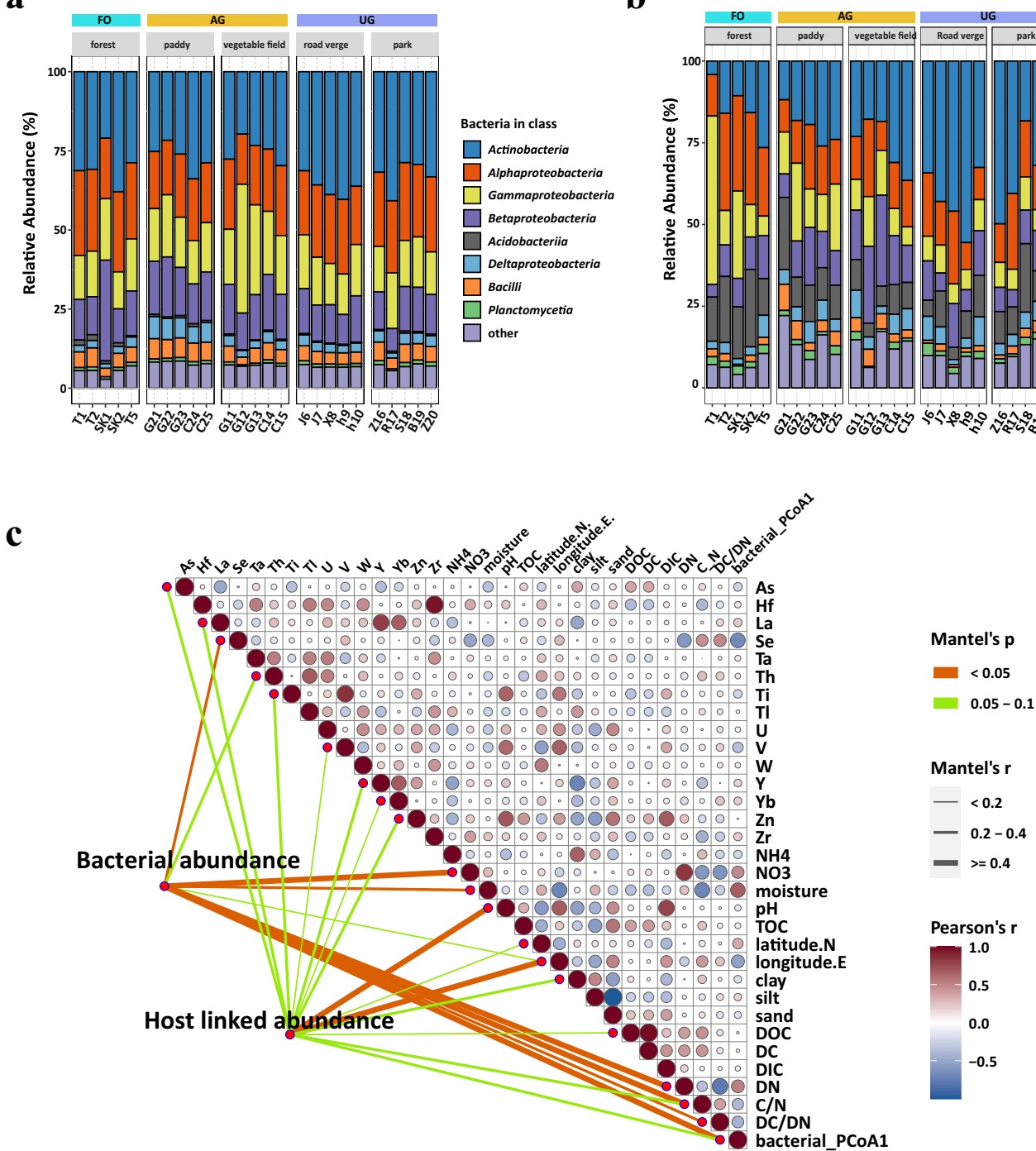

**Fig. 5 | The relationship between bacteria, host-linked viruses and environmental factors. a** Class distribution of bacterial compositions of metagenomes from various land use types. **b** Relative abundances of lineage-specific phages grouped at the host class level. **c** Mantel test showed the correlation between environmental factors and lineage-specific phages (top node, host-linked abundance) and bacterial community (bottom node, bacterial abundance). The size of circle represents Pearson Correlation Coefficient (*r*) among different environmental factors. The *p* values of mantel test were adjusted using FDR. The statistical test used was two-tailed. Source data are provided in the Source Data file.

12,314 vOTUs (20.6%) were classified as "regional". Only 2771 vOTUs (4.6%) were present in at least two zones and were defined as "multi-zonal", of which zone FO and AG only shared 103 vOTUs, whereas 1479 vOTUs were shared by zone FO and UG. In addition, only 76 vOTUs were detected in all zones, whereas there were 7398 bacterial species (64.4% of all) presented in all zones (Fig. 6a).

The microdiversity of vOTUs increased from local, regional to multi-zonal, despite that in FO, significant difference in the

microdiversity between regional and multi-zonal vOTUs was not observed (Wilcox test: *p* > 0.05; Fig. 6b). The microdiversity of multi-zonal vOTUs in FO were significantly lower than UG and AG (Wilcox test: *p* < 0.01; Fig. 6c). Furthermore, we also observed that the multi-zonal vOTUs have a significant higher popANI from intra zones compared with those from inter-zones (Supplementary Fig. 13).

The connection number of node (vOTU) in the gene-sharing network constructed by vConTACT2 were calculated. We found that

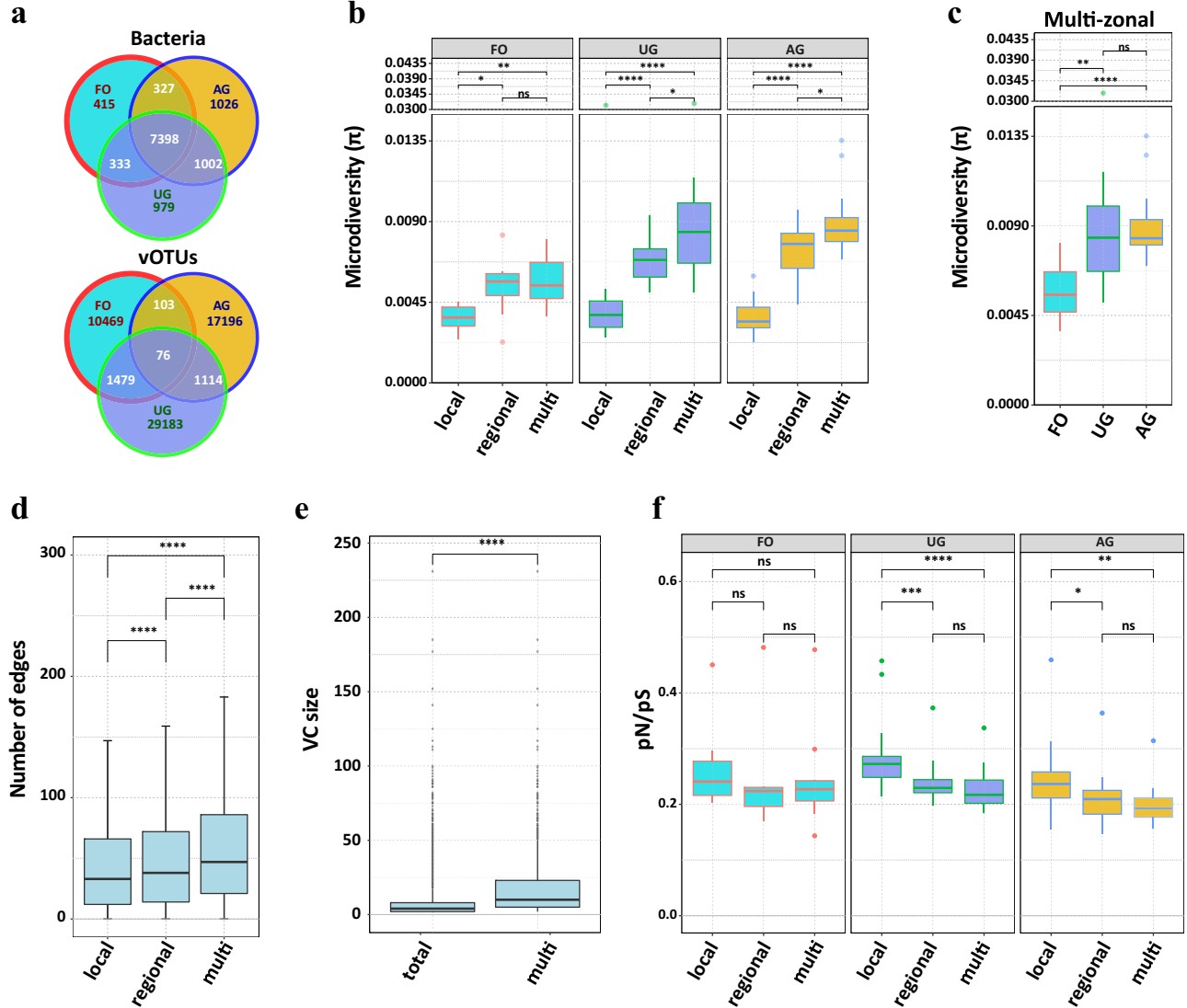

**Fig. 6 | Evolutions of soil vOTUs with different geographic ranges. a** Venn diagram showing the shared vOTUs (top) and bacterial species (bottom) among three land use zones. **b** The microdiversity of local, regional, and multi-zonal vOTUs in zones FO ($n = 10$), UG ($n = 20$) and AG ($n = 20$). **c** The microdiversity of multi-zonal vOTUs among three land use zones FO ($n = 10$), UG ($n = 20$) and AG ($n = 20$). **d** The connection number of local ($n = 44,534$), regional ($n = 12,314$), and multi-zonal ($n = 2771$) vOTUs in gene-sharing network. **e** The size of total VCs ($n = 6009$) and multi-zonal VCs ($n = 1416$). The exact $p$ value less than 2.2e-16. **f** The averaged pN/pS values of local, regional, and multi-zonal vOTUs in zones FO ($n = 10$), UG ($n = 20$), and AG ($n = 20$). For boxplots, the minima, maxima, center, bounds of box and whiskers in boxplots from bottom to top represented percentile 0, 10, 25, 50, 75, 90, and 100, respectively, the difference between different land use was tested using the Wilcox.test, ****, ***, **, *, and ns represents $p < 0.0001$, $p < 0.001$, $p < 0.01$, $p < 0.05$, and $p > 0.05$, respectively. The statistical test used was two-tailed. Source data are provided in the Source Data file.

the number of connections (median 47.00) of multi-zonal vOTUs with other vOTUs was significantly greater than that of regional one (median, 38; Wilcox test: $p < 0.001$), and regional one (medium, 38) was significantly greater than that of local (median, 31) (Fig. 6d). Subsequently, 2771 multi-zonal vOTUs were observed in 1416 VCs. The results showed that the size (median, 10) of VCs included multi-zonal vOTUs was significantly higher than the total (median, 4; Wilcox test: $p < 0.001$), the multi-zonal vOTUs prefer to exist in the large-sized VCs (Fig. 6e). These results demonstrated that the multi-zonal vOTUs shared genetic information with more vOTUs than local and regional in soil.

We identified genes whether under positive selection by evaluating the ratio of non-synonymous to synonymous mutations observed in gene sequences using the pN/pS equation. Of 726,331 genes tested from vOTUs with sufficient coverage (>5 mean coverage depth), 39,118 genes were identified as being under positive selection in at least one

sample, and almost all genes (39,027) were uniquely selected by certain region, of which 18,167 genes can be predicted with at least a function, and most of them were predicted responsible for structure and DNA metabolism (Supplementary Data 8). The frequency of each pfam number was counted, of which the top 10 positively selected proteins were dominated by those associated with viral structures, including 3981 genes related to Viral head morphogenesis (PF04233), 3101 genes for P22 coat protein (PF11651), and 695 genes for Phage virion morphogenesis (PF05069) (Supplementary Data 9). Uniquely positively selected genes were observed among the three zones, genes encoding DNA polymerase family A (PF00476) and Phage integrase family (PF00589) occupied the top two positively selected genes in zone FO, genes encoding Phage portal protein (PF04860) and Phage capsid family (PF05065) zone PT, and genes encoding Viral head morphogenesis (PF04233) and P22 coat protein (PF11651) were were positively selected in zone PV. The result was in accordance with that

the FO occupied higher putative lysogenic phages compared to other two zones (See source data). Higher pN/pS values were detected for the genes carried by local vOTUs than those carried by regional and multi-zonal vOTUs (Fig. 6f).

## Discussion

We conducted a comprehensive viromic examination of viral communities in soils various land use types. This size-fractioned viromic approach was far more prominent in the recovery of vOTUs than viral sequence mining from total soil metagenome[15]. A large volume of soil (300 g) was subjected to viral particles purification and viral DNA extraction, providing sufficient DNA amount for direct viromic library construction and sequencing that could reduce the bias due to the multiple displacement amplification (MDA) of low viral DNA yield. In addition to the extraction of eVLPs, we further adopted mitomycin C treatment of bacterial fraction to induce the release of iVLPs from their hosts. Sequencing and viromic analysis of eVLPs and iVLPs were conducted separately. The iVLPs and eVLPs viromes shared 11,467 vOTUs, accounting for only 19.2% of total assembled vOTUs. Implementing mitomycin-C treatment can recover more intracellular lysogenic viruses and proviruses (Fig. 4c, e) that are overlooked by the DNA-viromics approach mentioned previously and can advance our understanding of viral life strategies in different types of soils. The proviruses in viromes could originate from extracellular vesicle[31], nanobacteria[32], or lateral transduction[33] during which the proviruses induced by mitomycin-C could capture adjacent bacterial-host DNA. Through these efforts, we obtained, to our knowledge by far, the largest soil viral genomic catalog (LVD) consisting of 59,626 vOTUs, of which 8,112 vOTUs were recovered with complete genome, expanding ~3 fold of currently available complete viral genomes from soil metagenomes[34].

Most of the recovered vOTUs were classified into bacteriophages. The dominant families across all samples were *Siphoviridae*, *Myoviridae*, and *Podoviridae* (Supplementary Fig. 8), which was different from Antarctic soils including a fraction of nucleocytoplasmic large DNA virus (NCLDV)[16], mangrove sediment including ssDNA *Microviridae*, whereas consistent with those in permafrost[12]. Despite most of the recovered vOTUs can be taxonomically assigned to a family using a voting approach[35], over 95% of vOTUs obtained in our study did not clustered with reference genomes in the gene-sharing network, and only 93 vOTUs have been documented in IMG/VR v3 database[36], suggesting that this research cast light on part of the huge dark matter hidden in the soil, whereas more viral diversity remained uncovered.

The constructed LVD facilitated genome enabled analysis of macrodiversity and microdiversity of vOTUs across soils with various land use types. The structures of viral communities were significantly varied across land use zones (Fig. 2b), as well as bacterial community profiles (Supplementary Figure 9). The land use explained 38.4% of the variance of bacterial communities ($p < 0.0001$). Geographically distinct viral community compositions and structures have been well documented in natural ecosystems from global[21] to a very local scale within a 18 m² agricultural field[14]. We also compared the vOTUs in the LVD with those recovered from agricultural soil of Dezhou in our previous study[37], which locate in the North of China with more than 2000 km distance from Xiamen, only 23 vOTUs were shared between both datasets, indicating strong geographic isolation and existence of endemism for viral communities. Distance-decay was also observed with stronger spatial relationship in forest soil than urban and agricultural soils (Supplementary Fig. 6b), indicating geographic distribution patterns of soil viral communities. Nevertheless, in this study, the soil viral communities clustered according to their land use zones (Fig. 3b) and the samples within zones shared considerably more vOTUs compared to those shared among different zones (Fig. 3a), suggesting the anthropogenic land use change may dampen the viral

distance-decay relationship in soil. These results indicated the anthropogenic activities such as irrigation and fertilization exert a stronger niche selection pressure for soil viruses compared to natural factors, besides, the variation of vegetation types accompanied with land use changes would result in distinct rhizosphere and root detritusphere, thus similar types of land use could provide more alike niche for viruses and their hosts[38], and enhance their coevolution[39], resulting in different enriched host-linked viruses among various land uses (Supplementary Fig. 10). In addition, the significant correlation between viral and bacterial communities indicated shift in bacterial community structures induced by land use changes[40] contributed to the shift of viral communities (Mantel test $r = 0.24$, $p = 0.02$).

The dispersal potential of vOTUs was weaker than bacterial species, and the drift of vOTUs was rare among different habitats (Figs. 6a and 3d), which is consistent with previous studies[41]. The microdiversity of regional and multi-zonal vOTUs were significantly higher than local vOTUs (Fig. 6b), this trend is consistent with previous research which focused on global marine viral microdiversity in different ecological zones[9], suggesting the soil provided different niche selection pressure for local, regional and multi-zonal vOTUs[42]. Similar result was observed for multi-zonal vOTUs in zones AG and UG (Fig. 6c), revealing the niche in zone UG and AG provided a stronger selective pressure for multi-zonal vOTUs and driven viral differentiation[43]. The size of VCs containing multi-zonal vOTUs were the largest (Fig. 6d, e), further demonstrated that the multi-zonal vOTUs share genetic information with more vOTUs than local and regional vOTUs, suggesting the viral differentiation driven by niche-selection could increase viral speciation. Furthermore, the multi-zonal vOTUs presented a higher popANI value intra-zones compare with inter-zones (Supplementary Fig. 13) demonstrated that land use shaped differential evolution direction of viral lineage. Furthermore, genes of regional and multi-zonal vOTUs have a significantly lower pN/pS ratio than those of local vOTUs (Fig. 6f), indicating that soil viruses are likely under negative selection that could cause the extinction of most drifted vOTUs and thus leading to strong niche partition of soil viral communities when land use changes, thus only a small number of multi-zonal vOTUs with high microdiversity were remained and expanded their lineage to adapt the environmental change.

The infection dynamics of recovered vOTUs for specific host populations grouped at class level was assessed across three land use zones. A stronger correlation between the host-linked viral communities and host communities (Mantel test $r = 0.39$, $p < 1e-04$) was observed than that between viral communities and total bacterial communities. The compositions of bacterial communities and viral communities were similar across land use zones (Fig. 5a and Supplementary Fig. 9), however, the lineage-specific virus abundance significantly varied among habitats (Fig. 5b), with the abundances of major lineage-specific viruses correlating with geochemical parameters, particularly with soil pH. The abundances of *Actinobacteria*, *Alphaproteobacteria*, *Gammaproteobacteria*, and *Acidobacteria* remained relative stable across pH ranges, while their corresponding lineage-specific virus abundances significantly differed and correlated with pH (Supplementary Fig. 11). Actinophages were the most dominant bacteriophages in host-linked viral communities across land use zones (Fig. 5b), which has been reported as the largest identifiable group in aquatic ecosystems[44,45] but poorly studied in soil. Soil pH have been demonstrated as the major factor in shaping bacterial community compositions in global topsoil[2]. Consistent, the pH explained 27.9% of variance ($p < 0.0001$) of bacterial communities. Meanwhile, pH exhibited the strongest correlation with viral communities and host-linked viral communities compared with the other environmental factors (Fig. 3e), suggesting the vital role of pH in shaping the structure of viral communities and their host compositions[46]. These results imply that the impact of pH on the host compositions of viruses could contribute to the structuring of viral communities.

The relative abundance of putative lysogenic viruses increased from zone AG to UG to FO (Fig. 4a), indicating that the life strategies of viruses could be determined by land use types. Generally, the agricultural field bear the strongest stress of habitat disturbance from anthropogenic activities such as tillage, irrigation and fertilization, and that of the forest soil was the lowest. Both computational and experimental analysis demonstrated spatial structures select for lower virulence[47], with disturbed environment select lytic lifestyle partially due to the availability of susceptible host cells inhabited in spatial niches of soil for lytic infection[48,49]. Thus, land use change associated soil disturbance could be responsible for the decline of lysogenic viruses from forest soil to agricultural field. Significantly negative correlation between soil moisture and putative lysogenic viruses were presented (Fig. 4d). Generally, artificial irrigation in agricultural field and urban green space can result in higher soil moisture (Fig. 4d), which is often associated with more connectivity and dispersal as the water can fill the soil pore and promote the diffusion of viral-like particles in soils, and thus facilitating the movement of extracellular viruses and propagation of lytic viruses for new host cells[50,51]. In addition, DOC exhibited a significantly positive effect on the putative lysogenic virus abundance (Fig. 4d), indicating soil nutrient level is a major determinant for the viral shunt, which has been well documented in previous studies[25].

In conclusion, this study provides a database of high-quality viral genomes for a multitude analysis of soil virome, and a systematic investigation of patterns and drivers of soil viral macro- and microdiverisy. Land use types show a stronger effect on the speciation of soil viral communities than spatial distance, contrasting known biogeographic pattern. In addition, land use changes significantly shape the life strategy, host interaction and microdiversity of soil viral communities. Shift in the bacterial communities and geochemical factors, in particular, pH, moisture and nutrient level associated with land use changes, are the major determinants. With massive urbanization and changing climate, these advances provide insights into the response of soil viral communities to habitat disturbance and land management. Such understanding of soil viruses, and further interrogation of the linking between viral communities and complex ecosystem processes in soil, are critical for broader inclusion of viruses in ecosystem models in the Anthropocene.

## Methods

### Sample collection

Samples were collected from soil with various types of land uses, including forest, paddy field and vegetable field, urban park and road verge in Xiamen, China on Jul-26 2020, transported to lab on ice bag in a foam box, and stored immediately in 4 °C, which were further grouped into three land use zones as forest zone (FO), agricultural zone (AG), and urban green space zone (UG). These soils represent the major land use types, and are closely related to anthropogenic activities. Each land use type consisted of five randomly selected locations, and five replicate topsoil (0–20 cm) samples were collected and pooled into a composite sample from each location, resulting in a total of 25 soil samples (Supplementary Data 1 and Supplementary Fig. 1).

Soil samples were sieved (~2 mm) to remove stones for measure of soil properties. Soil pH (1:2.5 soil:water) was measured using a pH meter (IS126C Icon). Total carbon (TC), total nitrogen (TN), and total sulfur (TS) were determined with combustion method using a Vario MAX (elementar analysensysteme GmbH, Germany), and 49 metal elements were determined using an X-ray photoelectron spectrometry (Axis Supra, Kratos/Shimadzu). Soil moisture content was determined gravimetrically after oven dry at 60 °C with 12 h. The compositions of soil were detected using a laser particle size analyzer. The major soil components were sand and silt (Supplementary Data 1). The concentration of $NH_4^+$-N, $NO_3^-$-N, and $NO_2^-$-N were determined using an AA3 analyzer after extraction with 10-fold (weight/volume) of 2 M $CaCl_2$. The concentration of total organic carbon (TOC) (combustion method) and dissolved organic carbon (DOC) (0.5 M $K_2SO_4$ extracts) were detected using TOC-L CPH (Shimadzu, Japan).

### Soil DNA and viral DNA extraction

Soil viral DNA was extracted in accordance to a previous study[52] with some modification. Briefly, soil sample (300 × g) was extracted with PBS (900 ml) by shaking at 200 rpm 24 °C for 45 min, the supernatant was obtained by centrifugation at 3000 × g for 15 min at 4 °C, which was sequentially filtered through 5.0, 0.45, 0.22 μm cellulose membrane. The filtrate was used for the extraction of extracellular virus-like particles (eVLPs). To obtain intracellular VLPs (iVLPs), the membranes with microbes were transferred to 100 mL of PBS with 1 μg/mL of mitomycin C to induce the release of iVLPs from the hosts[53]. The mixture was filtered using a 0.22 μm cellulose membrane after incubation overnight (>8 h) at 150 rpm in dark under 30 °C[54]. The filtrates were continuously concentrated until ~250 μL solution using three 100 kDa Amicon Ultra centrifugal filter units (Millipore, American). The concentrated solution was treated with 20U DNase I (37 °C, 50 min) (Transgen Biotech). VLPs concentrates were filtered using a sterile 0.22 μm Millex-GP filter (Millipore, American) before viral DNA extraction using a TIANamp Virus DNA/RNA Kit (TIANGEN DP315, Beijing, China).

To extract total soil DNA for nontargeted metagenome sequencing, approximately 0.25 g soil was extracted with PowerSoil DNA isolation kit (Mo Bio Laboratories, Inc. Carlsbad, CA) according to the manufacturer's instructions. A NanoDrop ND-2000 spectrophotometer (NanoDrop, Wilmington, DE, USA) was used to determine the concentration and quality (A260/A280) of extracted DNA.

### Library construction, sequencing, and reads processing

Sequencing libraries were prepared using the ALFA-SEQ DNA Library Prep kit (mCHIP, China) following the manufacturer's recommendations and the index codes were added. Paired-end sequencing (150 bp) of total DNA and viral DNA were performed by MagiGene Co. Ltd. (Guangzhou in China) on the Illumina Novaseq 6000 platform respectively. The quality of raw sequences was assessed using FastQC v0.11.5[55]. Clean reads were obtained after adapter removing using cutadpat v2.11[56] with parameters: -a GATCGGAAGAGCACACGTCTGAACTCCAGTCAC -A AGATCGGAAGAGCGTCGTGTAGGGAAAGAGTGT, and quality filtering, trimming using trimmomatic v0.39[57] with parameters: LEADING:5 TRAILING:5 SLIDINGWINDOW:4:20 MINLEN:60.

### Analysis of metagenomes

The taxonomy of remaining high-quality reads were classified with Kraken2 v2.0.7-beta[58], the package Bracken[59] was used to estimate the relative abundances of detected taxa with normalization within a specific sample from Kraken2 classification results. The alpha- (Shannon's index) and beta- (Bray–Curtis dissimilarity) diversity of microbial community structures were performed using vegan[60] in R.

### Viral contig assembly, identification, and dereplication

The quality-filtered sequences of each individual viromes were assembled using metaspades v3.13.0[61]. Contigs larger than 1.5 kb were piped to VIBRANT v1.2.1[62] to predict putative viral contigs, meanwhile, the proviruses were identified as well. Contigs >10 kb or circular contigs <10 kb were obtained for further analysis. To remove the potential false positive viral contigs from VIBRANT, we applied a modified method proposed by Paez-Espino et al.[63]. Briefly, the ORFs of the putative viral contigs were predicted by prodigal v2.6.3[64] and were searched against the viral protein family (VPFs) dataset[21] containing 25,281 viral protein families using hmmsearch (e-value ≤ 1e-5). Putative viral contigs containing more than 3 ORFs with VPF hits were kept for further analysis.

Contigs were grouped into a viral operational taxonomic unit (vOTU) if they shared >95% nucleotide identity with more than 70% coverage of the shorter contigs using Perl script Cluster_genomes_5.1.pl (https://github.com/simroux/ClusterGenomes), we adopted an lower coverage threshold compare with the suggestion from Roux et al.[65] to accommodate more genetic variance and mosaicism of viral genomes in vOTUs. The longest contig was selected as representative contig of each vOTU. Subsequently, eVLPs and iVLPs datasets were merged and dereplicated at the population level to construct a total soil viral dataset across all viromes. The final dataset was named as LVD (land use virome dataset). The circular and putative temperate viral genomes in the LVD were identified through package VIBRANT[62]. The quality of genome was assessed using package VIBRANT and CheckV v0.7.0[34]. The lifestyle and genome quality of LVD information were listed in Supplementary Data 2. In the database LVD, 15.3% (9,172 vOTUs) lysogenic vOTUs were detected (Supplementary Data 3), and 4,844 (8.1%) completed genomes, 6,475 (10.8%) high-quality genomes and 15,156 medium-quality genomes (25.4%) were distributed into vOTUs through checkV (Supplementary Figure 2), only 133 genomes (0.2%) were identified as not-determined.

**Taxonomy assignment and host prediction of vOTUs**
vOTUs were clustered with viral genomes from NCBI RefSeq Release 201 using package vConTACT2[66], which assigned vOTUs to a known viral taxonomic genus and family if they were positioned with the viruses from Refseq in the same viral cluster[66]. For vOTUs that could not be assigned through vConTACT2, family-level taxonomic annotations were conducted using Demovir script (https://github.com/feargalr/Demovir) with default parameters and database[35]. The script search for homologies of proteins encoded by query viral contigs to the viral subset of the TrEMBL database[67], then the taxonomy of vOTUs were classified at family-level using a voting approach[35]. Additionally, we also checked the taxonomic assignment results between vConTACT2 and Demovir, which provided similar results at family-level with vConTACT2. We should note that the taxonomic framework adopted by the Demovir and vConTACT have some inconsistency with the ICTV 2021, such as the family *Siphoviridae*, *Podoviridae* and *Myoviridae* have been abolished in the latest release of ICTV. Here we presented the taxonomy assignment results from vConTACT2 and Demovir.

Microbial hosts of the representative contigs of each vOTU were predicted using VirHostMatcher-Net with short-contigs mode[68]. This software provides prediction of the prokaryotic host based on genomes of bacteria and archaea using previously developed CRISPRs (Clustered Regularly Interspaced Short Palindromic Repeats) and WlsH models[8,69]. The predicted host with the highest score and accuracy >90% was selected for further analysis.

**Estimation of the relative abundances of vOTUs**
An overview of the clean sequencing data of viromes for each site was provided in Supplementary Data 2. Since the minimum reads depth was 34 M, a threshold of 30 M reads was selected and the data was randomly subsampled without replacement across all viromes using Seqtk v1.3 (https://github.com/lh3/seqtk) to reduce the errors caused by sequencing depth. To estimate the relative abundance of vOTUs, virome reads were mapped to representative genomes of LVD dataset using bowtie2[70] with parameter --very-sensitive. The vOTUs were removed when less than 70% length of the representative contig was covered by reads mapped at a >= 95% identity, all reads mapped into remained vOTUs were used to calculate the RPKM (Reads Per Kilobase per Million mapped reads) value of each vOTU using package CoverM v0.5.0 (https://github.com/wwood/CoverM). For the Macrodiversity calculations, the RPKM values of each vOTU were normalized by total RPKM value per virome, which was used as a proxy for relative abundance. Each vOTU can be flagged by different features (e.g. lifestyle, host or VC) and the relative abundance with same flag were summed to

calculate the viral compositions with different features. The species accumulation curve was calculated through function Specaccum in R and the result indicated that the all vOTUs of LVD can be detected in subsampled viromes (Supplementary Fig. 3).

**Macrodiversity of viromes**
The alpha- (Shannon's index) and beta- (Bray–Curtis dissimilarity) diversity statistics of viral communities were performed using vegan in R[60]. The difference among viral communities among land use zones was evaluated using a PERMANOVA test (function "anosim" and "adonis") and the confidence intervals were plotted using function "ordiellipse" at the confidence limits of 95% and 97.5% using the standard deviation method.

The correlations between environmental variables and all the PCoA dimensions were evaluated using Mantel test (function mantel; permutations = 9999 and method = "spearman") after scaling (function scale) and calculating their distance matrices (function dist; method "bray" and na.rm = TRUE). The co-occurrence network among environmental factor, bacterial class and host-linked vOTUs at class level were generated using the Spearman Coefficient and was visualized by Gephi v0.9.2[71]. The relationship between the factors, pH and longitude, and the host-linked viral relative abundance at class level were assessed using linear model in R, respectively.

**Proportion of shared vOTUs between samples**
To identify shared vOTUs between zones or within a zone, the vOTU abundance table was transformed into a binary presence-absence matrix in R, where a relative abundance of 0.0001 was used as a threshold to determine the presence of a vOTU. The vOTU presence-absence data of paired extracellular and intracellular viromes were merged. The "proportion of shared vOTUs" that were present in different samples was calculated through the equation:

$$\text{Proportion of shared vOTUs} = ((Sn/a) + (Sn/b))/2 \qquad (1)$$

where "a" represents the numbers of vOTUs presented in one sample, "b" represents the number of vOTUs presented in another samples, and the "Sn" represented the numbers of shared vOTUs between the samples "a" and "b". The heatmap of the "proportion of shared vOTUs" was constructed in R v.4.0 using the package pheatmap, which was hierarchically clustered using method average. Wilcox.test was used to test the difference of "proportion of shared vOTUs" between different zones.

To explore the relative abundance of shared viral clusters (VCs) based on vConTACT2 among the different zones, we define the "all zone shared" as the VCs presented in all zones; "FO and AG shared" as the VCs only presented in zone FO and AG, "UG and AG shared" as the VCs only presented in zone UG and AG; "FO and UG shared" as the VCs only presented in zone FO and UG; "Unique" as the VCs exclusively presented in each zone.

**Classifying multi-zonal, regional, and local vOTUs**
vOTUs and bacterial species were evaluated for their distributions across the three zones of land use and plotted using the VennDiagram package in R. With reference to previous criteria[9], vOTUs were designated as "multi-zonal" if they were observed in >1 zone of land use, or zone-specific if they were observed in only one zone. Zone-specific vOTUs were further divided into local (observed only in 1 site) and regional (observed in ≥2 sites).

**Microdiversity of vOTUs**
The vOTUs with an RPKM > 5 across 70% of their representative genome in at least one sample in the datasets were flagged for subsequent microdiversity analyses. Nucleotide diversity (π) of flagged contigs, pN/pS for each gene of the flagged vOTUs were calculated using

package inStrain v1.5.3 according to Nei et al.[72]. The inStrain can identify and annotate biallelic and multiallelic SNVs and their frequencies at the positions where the quality-filtered reads differ from the reference sequence and where multiple bases are simultaneously detected at levels above the expected sequencing error rate[42]. Genes were considered under positive selection if pN/pS was >1. The popANI (population-level ANI) of multi-zonal vOTUs across all viromes were calculated using module compare of packages inStrains, the popANI were filtered if the population overlap <0.5 and the compared viromes come from same site.

To assess the microdiversity of vOTUs per zone, samples were randomly subsampled without replacement. Within each sample, π values of 10, 20, and 30 vOTUs of each distribution of zones (multi-zonal, regional, and local, respectively) were randomly selected and averaged. Within samples that lacked enough deeply-sequenced vOTUs, all the vOTUs were selected and averaged with particular range of land use. The subsampling referred to the methods proposed by Gregory et al.[9], different depths were chosen due to vOTUs with different geographic distribution occupied different proportion in LVD, of which local vOTUs were the most dominated, and followed by the regional vOTUs. Similarly, within each sample, pN/pS value of 10, 20, and 30 genes located on vOTUs with different distribution of zones (multi-zonal, regional, and local) were randomly selected and averaged respectively. This was repeated 1000x and the average of the all 1000 subsamplings was used as the final microdiversity and pN/pS value for each sample.

### Gene annotation

All genes were predicted using prodigal. ORFs were clustered at 95% identity over 70% contig length using CD-HIT v4.6[73] to reduce redundancy. The resultant ORFs were annotated by searching for matches against the InterPro protein signature database using InterProScan v5[74] with parameter -appl Pfam.

### Reporting summary

Further information on research design is available in the Nature Research Reporting Summary linked to this article.

## Data availability

The raw reads from Illumina viromes sequencing have been deposited in the NCBI under the project PRJNA691683. Raw reads from Illumina metagenomes sequencing were submitted to the NCBI under the project PRJNA746419. Clean reads from Illumina viromes sequencing were submitted to the ScienceDB (https://www.scidb.cn/s/yiIfAv). The database LVD have been deposited in figshare (10.6084/m9.figshare.19108391) [https://figshare.com/articles/dataset/LVD_v1_95-70_fna/19108391]. All bona-fide viral genomes of database LVD have been deposited in figshare (10.6084/m9.figshare.19740130) [https://figshare.com/articles/dataset/all_soil_phage_LVD_v1/19740130]. In additions, we also deposited another version of LVD which vOTUs clustered at thresholds of 95% average nucleotide identity over 85% alignment fraction using Perl script Cluster_genomes_5.1.pl into figshare (10.6084/m9.figshare.19743646) [https://figshare.com/articles/dataset/LVD_v1_95-85_fna/19743646]. Source data are provided with this paper.

## Code availability

The in-house Python scripts, R scripts, Perl scripts and relevant data used to generate figures of this study are publicly available on GitHub at https://github.com/liaohu1231/Virome.git (https://doi.org/10.5281/zenodo.7041414)[75].

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

## Acknowledgements

This study was financially supported by the National Natural Science Foundation of China (42177097 to H.L., 42021005 to Y.G.Z. and J.Q.S.) and the Alliance of International Science Organizations (ANSO-PA-2020-18 to Y.G.Z.).

## Author contributions

H.Liao, H.Li, and J.-Q.S. designed the experiment in-lab; H.Liao, H.Li, Q.-P.L., and X.-L.A. extracted metagenomic and viromic DNA; H.Liao processed all bioinformatic analysis; H.Li, J.-Q.S. and Y.-G.Z. applied the fund for this research; H.Liao and J.-Q.S. wrote the manuscript; Y.-G.Z., C.-S.D., X.-Y.Z. polished the manuscript. All authors read and approved the final manuscript.

## Competing interests

The authors declare no competing interests.
