## [Peer Review File · Nature Communications]

Reviewers' Comments:

Reviewer #1:

Remarks to the Author:

In this study, Liao and colleagues used a viromics approach to characterize the viral communities across different land-use sites in China. They found that (1) viral composition seems to be mainly structured by land use, (2) there appears to be a differential enrichment of lysogens across land-use zones, and (3) the intra-population micro-diversity of viruses shared across land-use zones is higher than the micro-diversity of spatially constrained viruses. The authors also implemented a mitomycin-C treatment to facilitate access to the diversity of integrated prophages.

Major comments

1. There are several missing details in the Methods section that are needed to properly assess the collection scheme and processing workflow followed in this study:

Were all samples collected on the same day? If not, did the authors test for correlations between collection time points and community structure?

What was the timeframe for processing the samples? Were they immediately processed or stored? If stored, under what conditions and for how long?

If samples were processed in different batches, were they randomized to avoid introducing confounding variation? If not, the authors need to test for potential batch effects.

2. There are a couple of discrepancies between the Bray-Curtis-based PCoA (Figure 2b) and the presence/absence-based hierarchical clustering (Figure 3b) results that should be addressed. First, I am curious as to why several viromes from all sample types seem to be converging at the center of the PCoA (Figure 2 - near the 0,-0.1 coordinates), and yet this pattern is not recapitulated in the hierarchical clustering dendrogram. Instead, there is a clear distinction by land-use zone based on the presence/absence-based hierarchical clustering dendrogram (Figure 3b). Second, it is also interesting that most forest samples separate from the rest of the viromes across the first axis of the PCoA, yet agricultural samples are the ones detected as outliers in the dendrogram. Given that differences between land-use zones is the central focus of this study, these differences should be further explored. I would suggest redoing the hierarchical clustering analysis based on Bray-Curtis dissimilarities and assessing if the patterns by land-use zones still hold. Additionally, it might be a good idea to update the presence/absence analysis with a more robust metric (like Jaccard) as the current method (the average between the percentages of shared VPs across pairs of samples) can be highly susceptible to outliers.

3. A substantial section of the paper is centered around the abundance patterns of lysogenic and lytic phages across samples. This focus makes sense as one of the most interesting aspects of this study is the use of mitomycin C to promote lysogen induction to explore the diversity of integrated prophages. However, the only evidence that the authors used to classify contigs as lysogenic is the *in silico* predictions generated by VIBRANT. I scanned the original publication of this bioinformatic tool and there is no clear explanation of how these annotations are generated; moreover, their GitHub repository only notes that these classifications are based on the presence of integrases or whether the VP was identified as integrated into a bacterial contig

(<https://github.com/AnantharamanLab/VIBRANT/issues/16>). This is a very big limitation and the language used to describe these results should reflect this. I would suggest fully explaining the biases of this approach in the Results section and referring to these VPs as “putative lysogens” rather than “lysogens”. This is important to avoid giving the impression that this approach is enough to identify lysogenic phages. Furthermore, the authors could try bolstering some of their claims by inspecting the metagenomes generated for each sample. For example, they could use VIBRANT to identify contigs with integrated prophages and see if there is an overlap with the VPs detected in the viromic samples (e.g. is there an enrichment of VPs identified as integrated into metagenomic bacterial contigs in the mitomycin-C treated viromes?).

4. In addition to the previous comment, I am confused by the use of subsampling in the comparisons between mitomycin treatments (Figure 4b). The libraries had originally been rarefied to an equal depth of 30 M reads so there’s no need to perform an additional subsampling. Furthermore, based on the number of replicates in Figure 4a, it seems that this approach was not followed for this related analysis. The authors need to redo the comparisons in Figure 4b without subsampling.

5. In the microdiversity analysis, the authors profiled the allelic variants across the set of reads mapped to individual VPs in each individual sample in order to assess their nucleotide diversity. They showed (1) that multi-zone VPs tend to be more diverse than regional and local VPs and (2) that forest VPs are less diverse than the other two land-zones. Yet, they use these results to claim that multi-zone VPs have “diverse origins” across land-use types (e.g. L43). This is not properly tested in the study and the authors need to be more cautious in their interpretation of these results. Potentially, the authors could calculate the ANI of multi-zone VPs to test if differences in nucleotide identity are bigger between land zones than within land zones. InStrain can perform these calculations with the “compare” module so it shouldn’t be too difficult to implement this approach.

6. It is not clear why the authors subsampled at different depths for multi-zonal, regional, and local VPs (L285). Isn’t the point of subsampling equalizing the numbers across all levels? Their authors need to elaborate on their rationale.

7. Several analyses in this paper involve multiple tests across several individual features (e.g. Mantel tests assessing the correlation of each edaphic property and community composition [Fig2c] and population abundances [Fig5c]), yet there is no indication that the results were corrected for multiple comparisons. Results need to be updated accordingly.

8. While I appreciate the amount of work that authors put into generating and analyzing their data, it is unfortunate that they didn’t spend more time polishing the manuscript to improve its readability. In addition to the more detailed description of some of these issues in the comments below, here’s a brief and non-exhaustive summary of things that could have been easily improved prior to submission that would have helped reviewers (and, eventually, readers) better appreciate the results described in this paper:

- Non-matching legends and figures (e.g. [1] the colors in Figure 1a don’t match the legend description, [2] the y-axes in Figure 4ab indicate relative abundances, yet the legend indicates these are proportions).
- Inconsistent use of color and labels across figures (e.g. [1] each relevant figure has a different

palette for land-use zones and soil types; [2] "road verge" and "treelawn" are used)

- Poor palette choices (e.g. it is very difficult to detect differences between colors in Figure 3b)
- Uneven text size across panels (e.g., the axis text size in Figure 6e is very small and difficult to read)
- Unnecessarily busy plots that are extremely difficult to parse and derive any insights from (Figure 5c).
- Referencing non-existent figures in the text (e.g.L385-L387)

The majority if not all of these issues should have been resolved prior to submission.

Minor comments

L37 - L39 - It is true that the main signal seems to be coming from land-use zones but I wouldn't discard geographic distance as an additional driver of community composition. There is still a lot of variation within soil types that could potentially be linked to geographic distance. For example, the relationships in the forest samples in Figure3b seem to be tracking spatial separation between sites. This could be easily tested by comparing spatial distance vs community dissimilarity within each soil type.

L118 - L121 - Based on this description, it is not clear what the collection scheme was. This sentence states that each of the five land use types consisted of five locations, and that for each location, five replicates were collected. Thus, there should be a total of 125 samples (5 land use types x 5 locations x 5 replicates), yet the authors only report 25. Is this because the 5 replicates per location were pooled together before DNA extractions? Was only one replicate used for viromics and the rest were reserved for other analyses (soil chemistry, total metagenomics, etc..)?

L138 - L146 - I am surprised by some of the amounts described in this section. Did authors use the entire ~900 mL supernatant to perform consecutive filtrations (presumably through syringe filters) and then concentrate into 250 uL with Amicon filters (which as far as I know have an upper limit capacity of 50mL)? Did they have to divide into subsamples (as described in <https://doi.org/10.1038/s43705-022-00110-x>) and then pooled the concentrated VLP suspensions? More detail is needed in order for readers to understand the extent of the lab work presented here.

L144 - Why did the authors decide to use this concentration of mitomycin C? Reference?

L159 - I couldn't find a description of this kit online. Was it tagmentation-based? If so, authors need to acknowledge the biases of such an approach (<https://doi.org/10.1186/1471-2164-14-320>)

L186 - L188 - Was this dereplication independent from the nucmer-based clustering described in L182 - L185? If so, what was the approach followed for this step?

L198 - L200 - This sentence was copied verbatim from 10.1016/j.chom.2019.09.009. Rephrase to avoid plagiarism

L214 - L217 - Was this filtering step performed at the read level (i.e., reads with less than 70%

coverage and <95% identity were removed prior to generating the count table) or at the VP level (i.e. after mapping reads, you removed all VP when less than 70% of the contig was covered by reads mapped at a <95% identity? The way that it's written seems to indicate the filtering was done at the read level but this step is typically performed at the VP level (see: <https://doi.org/10.7717/peerj.3817>)

L313 - Explain what eVLP and iVLP are. I know that this information is in the Methods section but making it accessible to the reader will improve the readability of the paper. Also, the Methods section in Nature Communications is at the end of the manuscript so defining these terms here is needed.

L320 - L323 - All of this information is not really used in the manuscript. I would recommend omitting this part and just referencing the supplementary table in the Methods section in case readers need additional information about the VPs.

L327 - Again, define what a VC is so readers don't have to parse through the Methods section.

L373 - Show the distribution of alpha diversity metrics across samples. You already did it for the bacterial communities (Figure S5b). why not for the viruses?

L378 - This is only a correlation, you cannot really say that pH is driving community composition. Rephrase

L385 - L387 - Figure 5S is not displaying the alpha-diversity trends of lysogenic phages, and no other figure matches this description.

L387 - What is the rationale behind performing a random sampling approach for Figure 4b but not Figure 4a? As I understand, a subsampling was already performed when the libraries were rarified to 30M reads prior to read mapping. Therefore, the samples should be readily comparable without performing any additional subsampling. The analysis needs to be updated.

L389 - L391. Were regression model assumptions (normality, homoscedasticity, ...) tested to make sure that linear models were appropriate to perform these tests?

L418 - What is DC/DN?

Figure 1 - The legend indicates there is a blue and a purple line, but the figure shows a green and pink line instead.

Figure 2 - Define the land zone use abbreviations in the legend to help the reader interpret the data.

Figure 3 - Some of the colors in the dendrogram of panel B are very similar and difficult to distinguish.

Figure 4 - Some of the facet names in panels a and b are different from the ones used in other figures. (e.g. "tree lawn" probably refers to "road verge"). The axes in Figure 4ab indicate that the boxplots are displaying the distribution of relative abundances but the legend suggests that they are actually

displaying proportions (i.e., how many contigs were classified as lysogenic). Clarify.

Figure 5 - It is very difficult to parse Figure 5c. I would recommend replacing it with a more informative plot. For example, a bar plot displaying the number of significant correlations detected across each edaphic property.

Style comments

L30 - Update "soil virome remained" to "soil virome remains"

L36 - Update "mitomycin" to "mitomycin C"

L60 - Update "pivotal in" to "pivotal to"

L67 - Update "lag" to "lags"

L119 - Update "was consisted" to "consisted"

L128 - L129. This is just gravimetric soil moisture. Simply describe what you did.

L175 - Update "qualified" to "quality-filtered"

L202 - Update "was predicted" to "were predicted"

L218 - Update "was normalized" to "were normalized"

L234 - Update "mantel test" to "Mantel test"

L258 - Update vContact2 to vConTACT2

L318 - Update "VPs were obtained with at least one complete viral genome" to "VPs genomes were classified as complete"

Reviewer #2:

Remarks to the Author:

Liao et al. generated paired viromic dataset for both intracellular and extracellular viruses extracted from forest, agricultural and urban soils to investigate the eco-evolutionary responses of soil viruses to land use changes. Land use type and environmental factors such as soil pH and moisture were found to potentially influence assemblages and life strategies of viral communities. Overall, the research question is important, and the empirical method is solid. My main concerns are 1) informatic methods of screening and clustering viral sequences and host assignment need more explanations (see comment 15, 16, 19, 20); 2) the manuscript is loaded with many analyses and results. More work is needed to better synthesize the results to support

clear take-home messages. More detailed comments are listed in the attachment. it is nice work and hope the comments can help to improve the clarity and credibility of the paper.

1. Line 40, host compositions instead of 'host ranges'?
2. Line 43, how the 'ubiquitous VPs' were defined? Unclassified? Please re-write the long sentence for clarity, 'Microdiversity...upon land use types'.
3. Line 66, missing reference for 'aquatic ecosystems' and 'mammalian guts'.
4. Line 67, 'yet' is incorrectly used in the sentence.
5. Line 82-83, 'bacteria are the hosts of phages that dominate soil virome'. That is true for DNA viromes. The recent soil RNA virome study (Hillary, Luke S., et al. "RNA-viromics reveals diverse communities of soil RNA viruses with the potential to affect grassland ecosystems across multiple trophic levels." ISME Communications 2.1 (2022): 1-10.) showed the predominance of Pisuciricota.
6. Line 119, how the locations were selected? Randomly? How far were the five locations from each other?
7. Line 129-130, two sentences?
8. Line 145, 'hosts'.
9. Line 146, instead of 'for', 'at' 150 rpm? under 30°C?
10. Line 148, '(Millipore, ?)'.
11. Line 164-165, please provides cut-offs.
12. Line 168-170, please separate it into two sentences and provide cut-offs so the others can repeat the procedures.
13. Line 169, were the relative abundances of the detected taxa normalized? If yes, please provide the details.
14. Line 179, missing version information of 'prodigal'.
15. Line 180, the screening criteria used in the reference 40 is different from what was described in this study. If the filter of 'more than 3 ORFs with VPF hits' is not published elsewhere, have the authors benchmark the method? It may be not the focus of this study, but it is the key to get a confident set of viral contigs for the downstream analysis.
16. Line 182-183, I would encourage the authors to refer to the clustering cutoffs ('standard thresholds of 95% average nucleotide identity over 85% alignment fraction') mentioned in Roux, Simon, et al. "Minimum information about an uncultivated virus genome (MIUViG)." Nature biotechnology 37.1 (2019): 29-37. As the clusters here is called 'viral population' that have been used by other relevant studies, it would be important to apply the same cut-offs. If the authors are not intended to compare to other studies, it

would be also acceptable to clearly define the clusters used in this study and acknowledge the differences from the 'viral population' used in other study.

0. Line 185, use 'contig' instead of 'genome'
1. Line 199, 'contigs in question'? maybe 'query contigs'?
2. Line 197, is Denivir a published or peer reviewed tool or method?
3. Line 202, have the authors checked the purity of the host assignment of each 'viral population'? were all the viral contigs within each 'viral population' assigned with one host taxonomy? I assume that the authors only used the representative contig to predict the host of the whole viral population. If not, please make it clear in the text.
4. Line 210, I would encourage the authors to include the sequencing statistics (e.g., total numbers of reads, numbers of quality-filtered reads, numbers of assemblies, N50, percent of reads that contribute to viral contigs, percent of reads that are non-viral...) in supplementals for studies using omic data.
5. Line 220. Please re-write the sentence 'Each VP can be flagged by different features' to make it clear.
6. Line 236-239, please double check the grammar and words used in this sentence.
7. Line 251, please consider adding quote to the variables that were described.
8. Line 268, a previous criterion, or previous criteria.
9. Line 334, 'vConTACT2'. Please double check the tool names were correctly spelled out throughout.
10. Figure 1a, 'blue line' in legend but green line in figure panel. Pink line was described as purple in the legend.
11. The legend says fig1c only included the taxonomy assignment by Demovir but both the vConTACT2 and Demovir results were described in the main text.
12. Line 345, 'sample sites' mean 'types of land use'? there is a mixed use of 'sample site', 'types of land use', 'zones', 'Types' (in figure 2b legend label), and 'samples'. I would encourage the authors to double check the manuscript and use these terms consistently.
13. How about the dissimilarity of intracellular VS extracellular viral communities for each land use type (if we have enough replicates to test it)? if the intracellular VS extracellular viral communities were more similar in samples with a certain land use type, it may suggest a relatively faster turnover of that virome? It can be related to the host/microbial activities? It would be nice to dig more into the iVPs and eVPs before combining them.
14. Line 358-360, are these results in one of the supplementals?
15. Figure 2c. no asterisks were shown in figure panel but described in legend. Please renumber the figure panels as the current fig2c were mentioned after fig3a-d in text.
16. Line 382, grammar check
17. How were the abundance estimates and community compositions of the mitomycin-C induced viral communities different from the informatically predicted lysogenic viruses (by VIBRANT)?
18. There are dark and light blue lines in figure 4c. I assume the line in light blue was for all the datapoints? Please make it clear in the legend label and text.
19. I would encourage the authors to use the same colors for FO, UG and AG across figures.

17. Line 482, in addition to improving the VLP recovery, implementing mitomycin-c treatment can recover the lysogenic viruses that are overlooked by the DNA-viromics approach mentioned previously and can advance our understanding of viral life strategies in different types of soils.
18. Line 505, grammar check.
19. Line 508, Paez-Espino, David, et al. "Uncovering Earth's virome." *Nature* 536.7617 (2016): 425-430.. this can be a better ref for viral research on a global scale.
20. Line 519-520, have the authors directly tested how the land use effect the bacterial community?
21. Line 527-542, please avoid repeating too many Results in Discussion.
22. Line 543-547, the authors could possibly test to what percent that soil pH can explain the variance observed in the bacterial communities or host communities. With support from data, these sentences will sound with less speculations.
23. Line 549, 'lysogenic viruses' here refer to the informatically predicted or mitomycin-c induced or both? (Related comment 34)
24. Line 555, 'creation of empty sites for lytic infection'? please re-write.
25. Line 561, please add a reference to support 'since higher...lytic viruses'.
26. Line 563-564, or 'viral shunt'?
27. The logic does not flow well from soil disturbance/land use to soil moisture. Please add more discussion or examples to connect the dots and make it easier for the readers. I would encourage the authors to re-organize the discussion. The discussion does not necessarily follow the same order of how the analyses were conducted but synthesizes the information from different aspects and avoid repeating too much from Results

Reviewer #3:

Remarks to the Author:

General comment for author:

The authors investigate the ecological pattern of soil viruses across different soil types, including forest, paddy, park, road verge, and vegetable field. The results showed that virus communities clustered based on land use and found pH in particular as the ecological driver. Furthermore, the author also examined the microdiversity of the virus communities and found evidence of strong speciation based on the land use. Below are my specific comments and suggestions:

I have an issue with the term "virus population" that the authors use throughout the manuscript system. The recommended cut-offs that were being proposed at 95% average nucleotide identity over 85% alignment fraction are for species-rank virus groups or virus operational taxonomic units (vOTU) (<http://dx.doi.org/10.1038/nbt.4306>), which is based on the reference genomes. To my knowledge, only the ocean system has evaluated how these $\geq 95\%$ ANI cutoff of the viral population definition is in nature (10.1016/j.cell.2019.03.040). Therefore, I suggest the authors use vOTU instead of "viral population". Otherwise, I would encourage the authors to do the same evaluation/validation for the soil system.

Results

Line 313-314: you mentioned removing "false positive". However, I could not find in the Methods section how you do it. Can you please explain it? If this is an integral/important part of your pipeline, please write it down in the Methods.

Line 315-316: "the iVLPs and eVLPs viromes shared 11,467 (19.2%)". Should you refer to this figure 3/6? If yes, it is a good idea to move this figure up.

Line 334-338: you compare the vConTACT2 and Demovir, yet you only show the figure for Demovir. Please make one for vConTACT2 as well and write "vConTACT2" consistently. See my

comment on figure 1C.

Line 343-351: Didn't you mention in the Methods about virome alpha diversity (Shannon's H)? Can you please show this result in the figure?

I do not understand why not showing beta diversity comparing sites/land use (like you showed in PCoA plot), instead of inter- and intra-site? I wonder what would the clustering look like when you use the NMDS plot?

I should say the result is quite interesting that the vOTUs are clustered based on the land use. I would also check if the Shannon's H of these three categories, whether they are statistically significant or not?

Line 372-373: can you please make a figure for Shannon's H and Simpson (or other diversity indexes)? I do not recall you mentioned Simpson in your Methods?

Line 387: here you said "alpha-diversity of lysogenic phage showed a significant difference among the three zones (Figure S5)", while referring to Figure S5 that shows microbial community PCoA and Shannon's H. Did you see any significant difference in Shannon's H between lytic and lysogenic per land use/sites?

Line 458-464: Can you explain more on these positively selected genes? What are niche defining genes? What genes are specific to certain soil samples/land use? This is the part supposed to get more coverage, since you have this in your title (eco-evolution).

Discussion

Many of the discussions are repetition of the results. This is not adding any substance to the discussion.

For example, line 467-487. Please check throughout the discussion.

Line 489: "Most of the recovered VPs were classified into dsDNA viruses and bacteriophages". What is the difference between dsDNA and bacteriophages? I don't recall Baltimore virus classification nor ICTV have dsDNA and bacteriophage differently. Or did you mean dsDNA viruses "or" bacteriophages?

Line 491-492: I think it is a good idea to compare viruses in this study and those in other soil types (from public databases, for example. IMG/Vr). How similar and different are these viruses?

Line 513-518: you explained why virus communities clustered according to land use rather than special proximity. However, I do not think your explanation is sufficient. Is there any biological explanation?

Line 566-583: I do not think you fully explain the importance of looking at microdiversity in viruses? Nor what does it mean?

Line 579: "increase the probability of possessing an adaptive genotype as land use changes". Can you show what these genotypes are? How important are they?

Methods

Line 189: you said, "temperate viral genomes in the LVD were identified through package VIBRANT". You also induced the prophages using Mitomycin C. My question is, did you assign all the iVLPs to be temperate phage? Are all the iVLP temperate phages? How much of the eVLP is a temperate phage? And should you refer this line to Supplementary table 2?

Line 190: "The quality of the genome was assessed using package VIBRANT and CheckV v0.7.0". Should you also refer this to Supplementary table 2?

Line 192-200: Taxonomy assignment. Although vConTACT2 does not intuitively give you a taxonomy assignment list. However, you can find the taxonomy information in "genome_by_genome_overview.csv" by checking "VC Status and VC Subcluster". So, my question is, can you explain more why not using vConTACT2 taxonomy results instead of using Demovir? And can you please check whether there is an agreement between the taxonomy assigned by "vConTACT2" and "Demovir"? if not how did you resolve the difference?

Line 209: "Estimation of viral population relative abundances". What is the virus length cutoff you used for this analysis? Previously (in line 177), you mentioned about "Contigs > 10 kb or circular contigs < 10 kb were obtained", but I am not sure whether you use this cutoff for all downstream analysis. Therefore, it would be useful to make it clear.

Figures

Figure 1A and 1B, I would personally put these two in supplementary. These are informative figures, but not really important to be put in the main manuscript.

Figure 1C, it is really hard to see the pie slices <10%. You can either 1) show the top 10 virus families, or 2) separate from "unassigned", you can combine those <10% as "Others".

Figure 3B, can you please put the coding/naming in figure caption.

Figure 4, this is a good figure. However, I think it would be helpful if you can show the alpha diversity of lysogenic across three land uses too.

Figure 5B, why did you show more bacterial-class-links for viruses in comparison to the bacterial composition in figure 5A? I would make two figures consistent and put the full list in the supplementary table.

Figure 5C, do the circle sizes mean anything?

Supplementary

Line 9-18: Should you put this explanation to Figure S1 caption?

Figure S4, I would explain in the figure caption the abbreviation of land use (i.e. FO, AG and UG) and name coding for x-axis.

Figure S5, can you also add the confidence intervals in the PCoA plot?

Figure S6, I would explain in the figure caption the abbreviation of land use (i.e. FO, AG and UG).

Supplementary table 2, please check the pdf when you convert from excel. There seems to be an issue with the converted table format.

Supplementary tables, it is advanceable for the authors to add a footnote on the content of the table. For example, supplementary table 3, what does it mean by Density, Internal weight, External weight, Quality? Please check all supplementary tables

Reviewer 1:

In this study, Liao and colleagues used a viromics approach to characterize the viral communities across different land-use sites in China. They found that (1) viral composition seems to be mainly structured by land use, (2) there appears to be a differential enrichment of lysogens across land-use zones, and (3) the intra-population micro-diversity of viruses shared across land-use zones is higher than the micro-diversity of spatially constrained viruses. The authors also implemented a mitomycin-C treatment to facilitate access to the diversity of integrated prophages.

Major comments

1. There are several missing details in the Methods section that are needed to properly assess the collection scheme and processing workflow followed in this study:

Were all samples collected on the same day? If not, did the authors test for correlations between collection time points and community structure?

What was the timeframe for processing the samples? Were they immediately processed or stored? If stored, under what conditions and for how long?

If samples were processed in different batches, were they randomized to avoid introducing confounding variation? If not, the authors need to test for potential batch effects.

Answer: All samples were collected on Jun-26 2020, transported to lab on ice bag in a foam box and stored immediately in 4 °C. Then these samples were immediately processed after the collection. Yes, the samples were randomly processed in different batches to avoid introducing variation. We have added the description in L117.

2. There are a couple of discrepancies between the Bray-Curtis-based PCoA (Figure 2b) and the presence/absence-based hierarchical clustering (Figure 3b) results that should be addressed. First, I am curious as to why several viromes from all sample types seem to be converging at the center of the PCoA (Figure 2 - near the 0,-0.1 coordinates), and

yet this pattern is not recapitulated in the hierarchical clustering dendrogram. Instead, there is a clear distinction by land-use zone based on the presence/absence-based hierarchical clustering dendrogram (Figure 3b). Second, it is also interesting that most forest samples separate from the rest of the viromes across the first axis of the PCoA, yet agricultural samples are the ones detected as outliers in the dendrogram. Given that differences between land-use zones is the central focus of this study, these differences should be further explored. I would suggest redoing the hierarchical clustering analysis based on Bray-Curtis dissimilarities and assessing if the patterns by land-use zones still hold. Additionally, it might be a good idea to update the presence/absence analysis with a more robust metric (like Jaccard) as the current method (the average between the percentages of shared VPs across pairs of samples) can be highly susceptible to outliers.

Answer: The PCoA based on the bray-distance and presence/absence-based hierarchical clustering based on percent of shared vOTUs between different viral communities, the big heterogeneity in soil viral community resulted in the low explanation in PCoA1 and PCoA2, thus several viromes from all sample types seem to be converging at the center of the PCoA, whereas the presence/absence-based hierarchical clustering ignored the variance in abundance, only considered the co-occurred vOTUs and some vOTUs with low abundance were removed, therefore the method highlighted the homogeneity between different viral communities. The figure 2b presented a clear distinction between different zones according to the results of anosim.test and adonis.test described in line 360-363, we also reanalyze the Bray-Curtis dissimilarities of viral community, and we updated the PCoA scatterplot (Figure 2b) based on PCoA1 and PCoA3. The result indicated a similar trend with the presence/absence-based hierarchical clustering. The sample T5 is the outlier of forest, and clustered into zone UG in PCoA and the presence/absence-based hierarchical clustering, rather than agricultural area. We also conducted the jaccard distance-based clustering, similarly, the samples were clustered according to land use types (see below), but different to the results of current method, the paddy and vegetable field in zone AG were separated by forest, although the paddy have a closer distance with

vegetable field compared with other types of land use, thus we would like to keep the current method.

The hierarchical cluster based on jaccard distance of viral communities.

3. A substantial section of the paper is centered around the abundance patterns of lysogenic and lytic phages across samples. This focus makes sense as one of the most interesting aspects of this study is the use of mitomycin C to promote lysogen induction to explore the diversity of integrated prophages. However, the only evidence that the authors used to classify contigs as lysogenic is the in silico predictions generated by VIBRANT. I scanned the original publication of this bioinformatic tool and there is no clear explanation of how these annotations are generated; moreover, their GitHub repository only notes that these classifications are based on the presence of integrases or whether the VP was identified as integrated into a bacterial contig (<https://github.com/AnantharamanLab/VIBRANT/issues/16>). This is a very big limitation and the language used to describe these results should reflect this. I would suggest fully explaining the biases of this approach in the Results section and referring to these VPs as “putative lysogens” rather than “lysogens”. This is important to avoid giving the impression that this approach is enough to identify lysogenic phages. Furthermore, the authors could try bolstering some of their claims by inspecting the metagenomes generated for each sample. For example, they could use VIBRANT to identify contigs with integrated prophages and see if there is an overlap with the VPs detected in the viromic samples (e.g. is there an enrichment of VPs identified as integrated into metagenomic bacterial contigs in the mitomycin-C treated viromes?).

Answer: Thank you very much for your professional comment. As your suggestion, we revised the description in the Results section, and adopted ‘putative lysogens’. Furthermore, we detected 525 prophages in database LVD, we also explored their relative abundance of prophages in viromes with different treatments. We found the mitomycin C can enhanced the recovery of prophages from viromes (see Figure 4e and

below). We have also added the description in L439-443.

4. In addition to the previous comment, I am confused by the use of subsampling in the comparisons between mitomycin treatments (Figure 4b). The libraries had originally been rarefied to an equal depth of 30 M reads so there's no need to perform an additional subsampling. Furthermore, based on the number of replicates in Figure 4a, it seems that this approach was not followed for this related analysis. The authors need to redo the comparisons in Figure 4b without subsampling.

Answer: Thanks for your comments. We reanalyzed the data without subsampling, the results showed significant difference between mitomycin-C treatments (Wilcox test, paired=TRUE $p < 0.001$) as well. The updated picture has been placed on Figure 4c.

5. In the microdiversity analysis, the authors profiled the allelic variants across the set of reads mapped to individual VPs in each individual sample in order to assess their nucleotide diversity. They showed (1) that multi-zone VPs tend to be more diverse than regional and local VPs and (2) that forest VPs are less diverse than the other two land-

zones. Yet, they use these results to claim that multi-zone VPs have “diverse origins” across land-use types (e.g. L43). This is not properly tested in the study and the authors need to be more cautious in their interpretation of these results. Potentially, the authors could calculate the ANI of multi-zone VPs to test if differences in nucleotide identity are bigger between land zones than within land zones. InStrain can perform these calculations with the “compare” module so it shouldn’t be too difficult to implement this approach.

Answer: Thank you very much for this professional comment. We calculated the conANI value of multi-zonal vOTUs using “compare” module in inStrain, the result (Figure S12 and below) indicated the differences in popANI of multi-zonal vOTUs are significant bigger between land zones than within land zones. Therefore, this statement was not supported by these results, we have revised relevant discussion in L582-601. We also added corresponding method description in L307-310.

6. It is not clear why the authors subsampled at different depths for multi-zonal, regional, and local VPs (L285). Isn't the point of subsampling equalizing the numbers across all levels? Their authors need to elaborate on their rationale.

Answer: We chose different depths for sub-sampling because the proportions of multi-zonal, regional, and local VPs in LVD was different, of which local VPs occupied the highest proportion, and followed by regional then multi-zonal VPs, thus we subsampled at different depths. This method referred to the description in a study on marine viral diversity (Gregory *et al.* 2019) at "Marine DNA Viral Macro- and Microdiversity from Pole to Pole". We have added the explanation in L316-320.

Reference: Gregory, A. C. *et al.* Marine DNA Viral Macro- and Microdiversity from Pole to Pole. *Cell* **177**, 1109-1123 e1114, doi:10.1016/j.cell.2019.03.040 (2019).

7. Several analyses in this paper involve multiple tests across several individual features (e.g. Mantel tests assessing the correlation of each edaphic property and community composition [Fig2c] and population abundances [Fig5c]), yet there is no indication that the results were corrected for multiple comparisons. Results need to be updated accordingly.

Answer: As your suggestions, we have updated the Fig 3e (from Fig2c) and Fig 5c after the *p* value were adjusted using FDR.

8. While I appreciate the amount of work that authors put into generating and analyzing their data, it is unfortunate that they didn't spend more time polishing the manuscript to improve its readability. In addition to the more detailed description of some of these issues in the comments below, here's a brief and non-exhaustive summary of things that could have been easily improved prior to submission that would have helped reviewers (and, eventually, readers) better appreciate the results described in this paper:

Answer: Thank you very much for these comments and we are really sorry for this and have carefully revised the manuscript to improve the readability.

- Non-matching legends and figures (e.g. [1] the colors in Figure 1a don't match the legend description, [2] the y-axes in Figure 4ab indicate relative abundances, yet the legend indicates these are proportions).

Answer: We have modified the color of FigS2 (previous Fig1a), and the y-axes of Fig 4ab.

- Inconsistent use of color and labels across figures (e.g. [1] each relevant figure has a different palette for land-use zones and soil types; [2] "road verge" and "treelawn" are used)

Answer: We have set a consistent palette for land use zones and soil types, and adopted road verge throughout the manuscript.

- Poor palette choices (e.g. it is very difficult to detect differences between colors in Figure 3b)

Answer: We have adopted a consistent and more readable palette throughout the manuscript.

- Uneven text size across panels (e.g., the axis text size in Figure 6e is very small and difficult to read)

- Unnecessarily busy plots that are extremely difficult to parse and derive any insights from (Figure 5c).

- Referencing non-existent figures in the text (e.g.L385-L387)

The majority if not all of these issues should have been resolved prior to submission.

Answer: Thanks for the comments. We have modified the palette and text size. Besides, we have removed some unrelated environmental factors in Figure 5c to improve the figure. Meanwhile, we also added the missing figure (alpha-diversity of putative lysogenic phage) in Figure 4b and below.

Minor comments

L37 - L39 - It is true that the main signal seems to be coming from land-use zones but I wouldn't discard geographic distance as an additional driver of community composition. There is still a lot of variation within soil types that could potentially be linked to geographic distance. For example, the relationships in the forest samples in Figure3b seem to be tracking spatial separation between sites. This could be easily tested by comparing spatial distance vs community dissimilarity within each soil type.

Answer: Thanks for the comment. We agree that the geographic distance as an

additional factor affecting the viral communities such as the longitude also presented significant correlation with viral communities (Figure 1c). Therefore, we have revised the sentence in L38-39 as ‘Significantly different profiles of viral communities were observed among land use types.’. As your suggestion, we observed geographic distance decay of viral communities within-zones (see below and Fig55b). A stronger distance-decay pattern was observed within zone FO compared with the other two zones, suggesting human activities may dampen the viral distance-decay relationship (see L407-409).

L118 - L121 - Based on this description, it is not clear what the collection scheme was. This sentence states that each of the five land use types consisted of five locations, and that for each location, five replicates were collected. Thus, there should be a total of 125 samples (5 land use types x 5 locations x 5 replicates), yet the authors only report 25. Is this because the 5 replicates per location were pooled together before DNA extractions? Was only one replicate used for viromics and the rest were reserved for other analyses (soil chemistry, total metagenomics, etc..)?

Answer: Sorry for the confusing description. The 5 replicates per location were pooled together before DNA extractions. We have added the description in L122.

L138 - L146 - I am surprised by some of the amounts described in this section. Did authors use the entire ~900 mL supernatant to perform consecutive filtrations (presumably through syringe filters) and then concentrate into 250 uL with Amicon filters (which as far as I know have an upper limit capacity of 50mL)? Did they have to divide into subsamples (as described in <https://doi.org/10.1038/s43705-022-00110-x>) and then pooled the concentrated VLP suspensions? More detail is needed in order for readers to understand the extent of the lab work presented here.

Answer. Yes. Actually, it was a tremendous amount of work for the concentration of supernatant. The filtrates were continuously concentrated until ~ 250 µL solution using three 100 kDa Amicon Ultra centrifugal filter units (Millipore, American) with approximate 20 cycles, 45 ml filtrates for each cycle. We have modified it in L149-152. Now we have developed a more efficient method to concentrate the VLPs before filtration to reduce lab work.

L144 - Why did the authors decide to use this concentration of mitomycin C? Reference?

Answer. The concentration of mitomycin C was adopted with reference to a previous study. We have added the reference in L148.

Reference: Liang, X. *et al.* Lysogenic reproductive strategies of viral communities vary with soil depth and are correlated with bacterial diversity. *Soil Biol Biochem* **144**, 107767, doi:10.1016/j.soilbio.2020.107767 (2020).

L159 - I couldn't find a description of this kit online. Was it tagmentation-based? If so, authors need to acknowledge the biases of such an approach (<https://doi.org/10.1186/1471-2164-14-320>).

Answer. The website (https://www.sohu.com/a/429714561_120729068) have a report with Chinese about the kit, and the website <http://www.mbiochip.com/list-47.html> have the description about the kit. The kit is not tagmentation-based according to the description.

L186 - L188 - Was this dereplication independent from the nucmer-based clustering described in L182 - L185? If so, what was the approach followed for this step?

Answer. The description in L184-187 could lead to misunderstanding. The identity and coverage were calculated based on the nucmer-aligning, and the clustering were conducted based on the output of nucmer, all the pipeline can be ran using the perl script Cluster_genomes_5.1.pl. We have modified the description in L196.

L198 - L200 - This sentence was copied verbatim from 10.1016/j.chom.2019.09.009.

Rephrase to avoid plagiarism

Answer. We have modified the description in L217-218.

L214 - L217 - Was this filtering step performed at the read level (i.e., reads with less than 70% coverage and <95% identity were removed prior to generating the count table) or at the VP level (i.e. after mapping reads, you removed all VP when less than 70% of the contig was covered by reads mapped at a <95% identity? The way that it's written seems to indicate the filtering was done at the read level but this step is typically performed at the VP level (see: <https://doi.org/10.7717/peerj.3817>)

Answer. Sorry for the confusing description. This step was conducted at the VP level. We removed all VPs when less than 70% of the contig was covered by reads mapped at a <95% identity. We have modified the description in L237-239.

L313 - Explain what eVLP and iVLP are. I know that this information is in the Methods section but making it accessible to the reader will improve the readability of the paper. Also, the Methods section in Nature Communications is at the end of the manuscript so defining these terms here is needed.

Answer. Thank you very much for your advice. We have modified it in L356.

L320 - L323 - All of this information is not really used in the manuscript. I would recommend omitting this part and just referencing the supplementary table in the Methods section in case readers need additional information about the VPs.

Answer. We have modified these sentences into the method sections in L203-208.

L327 - Again, define what a VC is so readers don't have to parse through the Methods section.

Answer. We have revised it in L365, and the VP have been modified as viral operational taxonomic units (vOTUs) according to the suggestion from another reviewer.

L373 - Show the distribution of alpha diversity metrics across samples. You already did it for the bacterial communities (Figure S5b). why not for the viruses?

Answer. Thanks for the comment. We added a figure (S6) to show the alpha diversity metrics of viral communities. The diversity of viral communities of three land use zones were comparable as indicated by Shannon (6.1–7.0) and Simpson (0.9878–0.9977) indexes (Figure S6).

L378 - This is only a correlation, you cannot really say that pH is driving community composition. Rephrase

Answer. Thanks. We have revised the sentence in L421-422.

L385 - L387 - Figure S5 is not displaying the alpha-diversity trends of lysogenic phages, and no other figure matches this description.

Answer. We have added the missing plot in Figure 4b to present the alpha-diversity trends of lysogenic phages (see below). We apologize for this.

L387 - What is the rationale behind performing a random sampling approach for Figure 4b but not Figure 4a? As I understand, a subsampling was already performed when the libraries were rarified to 30M reads prior to read mapping. Therefore, the samples should be readily comparable without performing any additional subsampling. The analysis needs to be updated.

Answer. Thank you very much for your suggestions. We have reanalysis the data without further subsampling, the updated plot has been placed on Figure 4c.

L389 - L391. Were regression model assumptions (normality, homoscedasticity, ...) tested to make sure that linear models were appropriate to perform these tests?

Answer. These models were test using normality through Q-Q plot and F-statistic using function summary(model) in R, and the p value of F-statistic have been added in the Figure 4d.

L418 - What is DC/DN?

Answer. The DC/DN represents dissolved carbon/dissolved nitrogen. We have added the description in L470.

Figure 1 - The legend indicates there is a blue and a purple line, but the figure shows a green and pink line instead.

Answer. Sorry for our carelessness. We have updated the figure in figure S2.

Figure 2 - Define the land zone use abbreviations in the legend to help the reader interpret the data.

Answer. Done

Figure 3 - Some of the colors in the dendrogram of panel B are very similar and difficult to distinguish.

Answer. We have updated the color of figure 3.

Figure 4 - Some of the facet names in panels a and b are different from the ones used in other figures. (e.g. “tree lawn” probably refers to “road verge”). The axes in Figure 4ab indicate that the boxplots are displaying the distribution of relative abundances but the legend suggests that they are actually displaying proportions (i.e., how many contigs were classified as lysogenic). Clarify.

Answer. We have updated the legend and caption as your comment.

Figure 5 - It is very difficult to parse Figure 5c. I would recommend replacing it with a more informative plot. For example, a bar plot displaying the number of significant correlations detected across each edaphic property.

Answer. Thank you very much for your suggestions. We have removed some unrelated environmental factors in Figure 5c to improve the figure.

Style comments

Answer. Thank you very much for these comments.

L30 - Update “soil virome remained” to “soil virome remains”.

Answer. Done

L36 - Update “mitomycin” to “mitomycin C”.

Answer. Done

L60 - Update “pivotal in” to “pivotal to”

Answer. Done

L67 - Update “lag” to “lags”

Answer. Done

L119 - Update “was consisted” to “consisted”

Answer. Done

L128 - L129. This is just gravimetric soil moisture. Simply describe what you did.

Answer. We have modified it in L132-133.

L175 - Update “qualified” to “quality-filtered”

Answer. Done

L202 - Update “was predicted” to “were predicted”

Answer. Done

L218 - Update “was normalized” to “were normalized”

Answer. Done

L234 - Update “mantel test” to “Mantel test”

Answer. Done

L258 - Update vContact2 to vConTACT2

Answer. Done

L318 - Update “VPs were obtained with at least one complete viral genome” to “VPs genomes were classified as complete”

Answer. Done

Reviewer 2:

Liao et al. generated paired viromic dataset for both intracellular and extracellular viruses extracted from forest, agricultural and urban soils to investigate the eco-evolutionary responses of soil viruses to land use changes. Land use type and environmental factors such as soil pH and moisture were found to potentially influence assemblages and life strategies of viral communities. Overall, the research question is important, and the empirical method is solid. My main concerns are 1) informatic methods of screening and clustering viral sequences and host assignment need more explanations (see comment 15, 16, 19, 20); 2) the manuscript is loaded with many analyses and results. More work is needed to better synthesize the results to support clear take-home messages. More detailed comments are listed in the attachment. It is nice work and hope the comments can help to improve the clarity and credibility of the paper.

Answer. Thanks for the comments, we have revised the manuscript according to your comments.

1. Line 40, host compositions instead of 'host ranges'?

Answer. Thanks for the comments, we have revised host ranges to host compositions throughout the manuscript.

2. Line 43, how the 'ubiquitous VPs' were defined? Unclassified? Please re-write the long sentence for clarity, 'Microdiversity...upon land use types'.

Answer. Here ubiquitous VPs refer to multi-zonal VPs, we have re-written the sentence to clarify in L42-44

3. Line 66, missing reference for 'aquatic ecosystems' and 'mammalian guts'.

Answer. We have added the references in L66-67.

4. Line 67, 'yet' is incorrectly used in the sentence.

Answer. We have revised it in L67.

5. Line 82-83, 'bacteria are the hosts of phages that dominate soil virome'. That is true for DNA viromes. The recent soil RNA virome study (Hillary, Luke S., et al. "RNA-viromics reveals diverse communities of soil RNA viruses with the potential to affect grassland ecosystems across multiple trophic levels." ISME Communications 2.1 (2022): 1-10.) showed the predominance of Pisuciricota.

Answer: Thank you very much for your suggestion. We have modified the description to DNA virome in L84.

6. Line 119, how the locations were selected? Randomly? How far were the five locations from each other?

Answer: We have added the description in L121. The locations were randomly selected for different types of land uses. The geographic positions of each location have been provided in Table S1, and the distance between them have been provided in Figure S5.

7. Line 129-130, two sentences?

Answer: We have revised it in L133-134.

8. Line 145, 'hosts'.

Answer: Done

9. Line 146, instead of 'for', 'at' 150 rpm? under 30 ?

Answer: We have revised it in L149.

10. Line 148, '(Millipore, ?)'.

Answer: Yes. We have revised it in L151.

11. Line 164-165, please provides cut-offs.

Answer: The cut-offs as LEADING:5 TRAILING:5 SLIDINGWINDOW:4:20 MINLEN:60.
We have revised it in L170-171.

12. Line 168-170, please separate it into two sentences and provide cut-offs so the others can repeat the procedures.

Answer: Revised as your suggestions (L169-171).

13. Line 169, were the relative abundances of the detected taxa normalized? If yes, please provide the details.

Answer: Yes, the relative abundances of taxa were normalized using the output of Bracken. We have added the detail it in L177-179 as 'the package Bracken³⁸ was used to estimate the relative abundances of detected taxa with normalization within a specific sample from Kraken2 classification results.'

14. Line 179, missing version information of 'prodigal'.

Answer: We have added the version of prodigal in L187.

15. Line 180, the screening criteria used in the reference 40 is different from what was described in this study. If the filter of 'more than 3 ORFs with VPF hits' is not published elsewhere, have the authors benchmark the method? It may be not the focus of this study, but it is the key to get a confident set of viral contigs for the downstream analysis.

Answer: Thank you very much for your suggestion. In our research, we provided the relatively loose criteria in this step due to three points: (1) The reference 40 adopted more than 5 ORFs with VPFs to identify the viral contigs recovery from metagenomes, whereas our contigs were assembled from viromes, most of bacterial contigs have been removed in lab. (2) The contigs were identified using the package VIBRANT which used database VOG, pfam, and KOALA-KOFAM to annotate viral marker genes, the VIBRANT can provide a high confidence for viral prediction and have been applied in many research to identify viral contigs such as Rambo *et al.*; and the origin article of VIBRANT have been cited more than 100 times since published in 2020. (3) The viral

genomic quality of identified viral contigs were furtherly assessed, only 133 genomes were identified as not-determined. Therefore, the criteria can remove most of false-positive genomes from VIBRANT.

Reference: Rambo *et al.* Genomes of six viruses that infect Asgard archaea from deep-sea sediments. *Nature Microbiology*, 2022

16. Line 182-183, I would encourage the authors to refer to the clustering cutoffs ('standard thresholds of 95% average nucleotide identity over 85% alignment fraction') mentioned in Roux, Simon, et al. "Minimum information about an uncultivated virus genome (MIUViG)." *Nature biotechnology* 37.1 (2019): 29-37. As the clusters here is called 'viral population' that have been used by other relevant studies, it would be important to apply the same cut-offs. If the authors are not intended to compare to other studies, it would be also acceptable to clearly define the clusters used in this study and acknowledge the differences from the 'viral population' used in other study.

Answer. We also realized the cutoff of 95% average nucleotide identity over 85% alignment fraction for cluster of viral genomes have been used in many other researches. There were four reasons for selecting 70% coverage as threshold, see below. We agree it is important to use the same cutoffs for the term "viral population". Thus we have modified the viral populations as vOTUs (viral operational taxonomic units) in the revised manuscript according to your comment and the suggestion from another reviewer.

1. In general, the cutoff was used to distinguish the species in bacteria and archaea such as 95% ANI, but the viral genomes have a higher mutant rate and mosaicism (horizon gene transfer) compared with bacterial and archaeal genomes. Therefore, in this research, we intended to used a relatively looser cut-off of coverage to tolerate more mosaicism and genetic variance of viral genomes and cluster more genomes in same viral cluster. We also provide all viral genomes and vOTUs clustered by standard thresholds of 95% average nucleotide identity over 85% alignment fraction into website figshare (61106 vOTUs) if other researcher needs these datasets (L339-343).

2. The 70% coverage were selected for keep the consistency with the mapped reads need to cover >70% length of vOTUs when we calculate the relative abundance of vOTUs.

3. We compared our vOTUs with those from our another published Dezhou soil virome with the same cutoff in this research.

4. We found the 95% identity have been applied as a general threshold in many previous researches, but the coverage threshold is not unified, such as the [doi:10.1016/j.cell.2019.03.040](https://doi.org/10.1016/j.cell.2019.03.040) using 80%, and <https://doi.org/10.1016/j.isci.2022.104418> using 75%.

Reference: Gregory, A. C. *et al.* Marine DNA Viral Macro- and Microdiversity from Pole to Pole. *Cell* **177**, 1109-1123 e1114, [doi:10.1016/j.cell.2019.03.040](https://doi.org/10.1016/j.cell.2019.03.040) (2019).

Li *et al.* A catalog of 48,425 nonredundant viruses from oral metagenomes expands the horizon of the human oral virome. *iScience*. doi.org/10.1016/j.isci.2022.104418 (2022)

17. Line 185, use 'contig' instead of 'genome

Answer: Done

18. Line 199, 'contigs in question'? maybe 'query contigs'?

Answer: Yes, we have revised the sentence it in L217.

19. Line 197, is Denivir a published or peer reviewed tool or method?

Answer: We used the tool Demovir to predict viral taxonomy in family-level according to description of the paper "The Human Gut Virome Is Highly Diverse, Stable, and Individual Specific" published in *Cell host & Microbe* in 2019.

20. Line 202, have the authors checked the purity of the host assignment of each 'viral population'? were all the viral contigs within each 'viral population' assigned with one host taxonomy? I assume that the authors only used the representative contig to

predict the host of the whole viral population. If not, please make it clear in the text.

Answer: Thank you very much for your professional suggestions. Yes, we did not check the purity of each “viral population”, and only the representative contig of each “viral populations” was used to predict their host. We have clarified this in L223.

21. Line 210, I would encourage the authors to include the sequencing statistics (e.g., total numbers of reads, numbers of quality-filtered reads, numbers of assemblies, N50, percent of reads that contribute to viral contigs, percent of reads that are non-viral...) in supplementals for studies using omic data.

Answer: We have added the information in Table S2 (e.g., numbers of quality-filtered reads, numbers of assemblies, N50, percent of reads that contribute to viral contigs).

22. Line 220. Please re-write the sentence ‘Each VP can be flagged by different features’ to make it clear.

Answer: We have revised the sentence to ‘Each vOTU can be flagged by different features (e.g. lifestyle, host or VC) and the relative abundance with same flag were summed to calculate the viral compositions with different features’ in L243-245.

23. Line 236-239, please double check the grammar and words used in this sentence.

Answer: We have revised the sentence it in L261-263.

24. Line 251, please consider adding quote to the variables that were described.

Answer: Thank you very much for your suggestions. We have revised it in L275-281.

25. Line 268, a previous criterion, or previous criteria.

Answer: Thank you very much for your suggestions. We have revised it in L293.

26. Line 334, ‘vConTACT2’. Please double check the tool names were correctly spelled out throughout.

Answer: Thanks for the comment. We have revised it throughout the manuscript.

27. Figure 1a, 'blue line' in legend but green line in figure panel. Pink line was described as purple in the legend.

Answer: Sorry for this, we have revised Figure S2 and also checked the other figures.

28. The legend says fig1c only included the taxonomy assignment by Demovir but both the vConTACT2 and Demovir results were described in the main text.

Answer: Thank you very much for your suggestions. We have supplemented Figure 1b presenting the taxonomy assignment results from vConTACT2.

29. Line 345, 'sample sites' mean 'types of land use'? there is a mixed use of 'sample site', 'types of land use', 'zones', 'Types' (in figure 2b legend label), and 'samples'. I would encourage the authors to double check the manuscript and use these terms consistently.

Answer: It does not mean "types of land use", we have revised it as "sites", and the terms "sites", "types of land use" and "zones" represents different groups of samples, for example, the sites represent the paired mitomycin C and non-mitomycin C treatment sample from same site. We have revised it in L386 and Figure 2b, and we have added the group information in Table S2.

30. How about the dissimilarity of intracellular VS extracellular viral communities for each land use type (if we have enough replicates to test it)? if the intracellular VS extracellular viral communities were more similar in samples with a certain land use type, it may suggest a relatively faster turnover of that virome? It can be related to the host/microbial activities? It would be nice to dig more into the iVPs and eVPs before combining them.

Answer: This is an excellent idea, thank you very much. We evaluated the dissimilarity between iVLPs and eVLPs of each land use type, significant difference was not observed between different land use types (see below). Therefore, we have not added the figure in this paper.

31. Line 358-360, are these results in one of the supplementals?

Answer: We have supplemented the results in Table S5.

32. Figure 2c. no asterisks were shown in figure panel but described in legend. Please renumber the figure panels as the current fig2c were mentioned after fig3a-d in text.

Answer: We have modified the Figure 2c as Figure 3e.

33. Line 382, grammar check

Answer: We have revised it in L428-429.

34. How were the abundance estimates and community compositions of the mitomycin-C induced viral communities different from the informatically predicted lysogenic viruses (by VIBRANT)?

Answer: A very good point, thank you very much. We actually intended to conduct such an analysis, while, although mitomycin C treatment would induce the release of prophage from host cell, the extracted iVLPs also include abundant lytic phages

released during the process. The VIBRANT can identify lysogenic phage based on the presence of integrase or whether the VP was identified as integrated fragment on a bacterial contig, however, a considerable proportion of lysogenic phages may remained unidentified due the limitation of current viral database. Therefore, we decided not to compare the results from the two approaches.

35. There are dark and light blue lines in figure 4c. I assume the line in light blue was for all the datapoints? Please make it clear in the legend label and text.

Answer: Sorry for the confusing figure. We have modified the figure and the legend.

36. I would encourage the authors to use the same colors for FO, UG and AG across figures.

Answer: As your suggestion, we have used the same color for FO, UG and AG across all figures.

37. Line 482, in addition to improving the VLP recovery, implementing mitomycin-c treatment can recover the lysogenic viruses that are overlooked by the DNA-viromics approach mentioned previously and can advance our understanding of viral life strategies in different types of soils.

Answer: Thank you very much for the excellent point, which have been supplemented in the text of the revised manuscript. We have replaced the sentence in L538-542.

38. Line 505, grammar check.

Answer: Done.

39. Line 508, Paez-Espino, David, et al. "Uncovering Earth's virome." Nature 536.7617 (2016): 425-430. this can be a better ref for viral research on a global scale.

Answer: Thank you. this reference has been cited in the revised MS in L564.

40. Line 519-520, have the authors directly tested how the land use effect the bacterial

community?

Answer: Yes, significantly different bacterial communities was observed across land use types, the adonis result indicated the land use types explained 31.3% of the variance. We have added the description in L561.

41. Line 527-542, please avoid repeating too many Results in Discussion.

Answer: Thank you very much for your suggestion. We have removed many unnecessary repetitions in this paragraph.

42. Line 543-547, the authors could possibly test to what percent that soil pH can explain the variance observed in the bacterial communities or host communities. With support from data, these sentences will sound with less speculations.

Answer: Thank you very much for your suggestion. We evaluated the contribution of pH to the variation of bacterial communities through adonis analysis. We have added it in L621.

43. Line 549, 'lysogenic viruses' here refer to the informatically predicted or mitomycin-c induced or both? (Related comment 34)

Answer: Here it referred to informatically predicted lysogenic viruses via VIBRANT.

44. Line 555, 'creation of empty sites for lytic infection'? please re-write.

Answer: We have re-written it in L637-642.

45. Line 561, please add a reference to support 'since higher...lytic viruses'.

Answer: We have added a reference in L639-644.

46. Line 563-564, or 'viral shunt'?

Answer: Yes, we have modified it in L646.

47. The logic does not flow well from soil disturbance/land use to soil moisture. Please

add more discussion or examples to connect the dots and make it easier for the readers.

Answer: Thank you very much for your suggestion. We have modified the logic from soil disturbance/land use to soil moisture in L643-646.

48. I would encourage the authors to re-organize the discussion. The discussion does not necessarily follow the same order of how the analyses were conducted but synthesizes the information from different aspects and avoid repeating too much from Results.

Answer: Thank you very much for your comment. We have removed some duplicated sentence, and we have reorganized some results and synthesized them such as the effect of pH, and moved the paragraph about microdiversity to follow the macrodiversity.

Reviewer 3:

General comment for author:

The authors investigate the ecological pattern of soil viruses across different soil types, including forest, paddy, park, road verge, and vegetable field. The results showed that virus communities clustered based on land use and found pH in particular as the ecological driver. Furthermore, the author also examined the microdiversity of the virus communities and found evidence of strong speciation based on the land use.

Below are my specific comments and suggestions:

I have an issue with the term “virus population” that the authors use throughout the manuscript system. The recommended cut-offs that were being proposed at 95% average nucleotide identity over 85% alignment fraction are for species-rank virus groups or virus operational taxonomic units (vOTU) (<http://dx.doi.org/10.1038/nbt.4306>), which is based on the reference genomes. To my knowledge, only the ocean system has evaluated how these $\geq 95\%$ ANI cutoff of the viral population definition is in nature ([10.1016/j.cell.2019.03.040](https://doi.org/10.1016/j.cell.2019.03.040)). Therefore, I suggest the authors use vOTU instead of “viral population”. Otherwise, I would encourage the authors to do the same evaluation/validation for the soil system.

Answer: Thank you very much for your professional suggestion. We have revised the term “viral population” as “vOTU” across the whole manuscript. We also realized that the cutoff of 95% average nucleotide identity over 85% alignment fraction for species-rank virus groups have been used in other researches. In general, the cutoff was used to distinguish the species in bacteria and archaea such as 95% ANI, but the viral genomes have a higher mutant rate and mosaicism (horizon gene transfer) compared with bacterial and archaeal genomes. Therefore, in this research, we adopted a less stringent criterion of coverage. We also provided all viral genomes and vOTUs clustered at 95% average nucleotide identity over 85% alignment fraction into website figshare (61,106 vOTUs) if other researcher needs these datasets.

Results

1.Line 313-314: you mentioned removing “false positive”. However, I could not find in the Methods section how you do it. Can you please explain it? If this is an integral/important part of your pipeline, please write it down in the Methods.

Answer: Thanks for the comment. The method was presented in the M&M section, we have revised it to make it more clear in L185.

2.Line 315-316: “the iVLPs and eVLPs viromes shared 11,467 (19.2%)”. Should you refer to this figure 3/6? If yes, it is a good idea to move this figure up.

Answer: Thanks for the comment. The results was only presented in numbers, not in any figure.

3.Line 334-338: you compare the vCONTACT2 and Demovir, yet you only show the figure for Demovir. Please make one for vCONTACT2 as well and write “vCONTACT2” consistently. See my comment on figure 1C.

Answer: The results from vCONTACT2 was presented in figure 1b in the revised MS, and we have revised “vCONTACT2” across the manuscript.

4.Line 343-351: Didn’t you mention in the Methods about virome alpha diversity (Shannon’s H)? Can you please show this result in the figure?

Answer: Thanks for the useful suggestion. We have added it in figure S6.

5.I do not understand why not showing beta diversity comparing sites/land use (like you showed in PCoA plot), instead of inter- and intra-site? I wonder what would the clustering look like when you use the NMDS plot? I should say the result is quite interesting that the vOTUs are clustered based on the land use. I would also check if the Shannon’s H of these three categories, whether they are statistically significant or not?

Answer: The alpha diversity of viromes have been supplemented in Figure S6.

Most of the bray-curtis dissimilarity values between different sites/land use types were close to 1 due to the strong heterogeneity of soil viral communities, thus a box plot showing the beta-diversity comparing sites/land use may not be informative for readers.

The NMDS plot showed a similar clustering pattern with PCoA as shown below. However, the Shannon's H did not present a significant difference between different zones (see Figure S6).

6. Line 372-373: can you please make a figure for Shannon's H and Simpson (or other diversity indexes)? I do not recall you mentioned Simpson in your Methods?

Answer: As your suggestion, the alpha diversity of viromes have been supplemented in Figure S6.

7. Line 387: here you said "alpha-diversity of lysogenic phage showed a significant difference among the three zones (Figure S5)", while referring to Figure S5 that shows microbial community PCoA and Shannon's H. Did you see any significant difference in

Shannon's H between lytic and lysogenic per land use/sites?

Answer: Sorry for the negligence. The alpha-diversity of putative lysogenic phage showed a significant difference among the three zones (see L436), and the alpha diversity of viromes have been added as Figure 4b (see below).

8. Line 458-464: Can you explain more on these positively selected genes? What are niche defining genes? What genes are specific to certain soil samples/land use? This is the part supposed to get more coverage, since you have this in your title (eco-evolution).

Answer: Thank you very much for your professional suggestion. We have modified this

paragraph (L517-524), added more details for positive selective genes such as their major pfam number and functional description.

Discussion

1. Many of the discussions are repetition of the results. This is not adding any substance to the discussion.

For example, line 467-487. Please check throughout the discussion.

Answer: Thank you very much for your helpful suggestion. We have checked throughout the discussion and removed unnecessary results.

2. Line 489: "Most of the recovered VPs were classified into dsDNA viruses and bacteriophages". What is the difference between dsDNA and bacteriophages? I don't recall Baltimore virus classification nor ICTV have dsDNA and bacteriophage differently. Or did you mean dsDNA viruses "or" bacteriophages?

Answer: Thank you very much for your professional suggestion. We removed the dsDNA viruses in the sentence in L547.

3. Line 491-492: I think it is a good idea to compare viruses in this study and those in other soil types (from public databases, for example. IMG/Vr). How similar and different are these viruses?

Answer: We agree this is a good idea. Actually, we have compared the database LVD and our published Dezhou viromes in L566-567. We also added the comparison between IMG/VR v3.0 and LVD, only 93 vOTUs in LVD shared more than 95% identity and 70% coverage with viral genomes in IMG/VR v3.0 in L554.

4. Line 513-518: you explained why virus communities clustered according to land use rather than special proximity. However, I do not think your explanation is sufficient. Is there any biological explanation?

Answer: We have modified the description in L573-574 as "These results indicated a

stronger effect of land use types on the dynamic of soil viral communities than spatial distance.”, we also think the previous explanation need more evidences, thus we have revised it.

5. Line 566-583: I do not think you fully explain the importance of looking at microdiversity in viruses? Nor what does it mean?

Answer: We have modified the description in L584-603. 'The microdiversity of regional and multi-zonal vOTUs were significantly higher than local vOTUs (**Figure 6b**), this trend is consistent with previous research which focused on global marine viral microdiversity in different ecological zones⁹, suggesting the soil provided different niche selection pressure for local, regional and multi-zonal vOTUs⁵⁴. Similar result was observed for multi-zonal vOTUs in zones AG and UG (Figure 6c), revealing the niche in zone UG and AG provided a stronger selective pressure for multi-zonal vOTUs and driven viral differentiation⁶¹. The size of VCs containing multi-zonal vOTUs were the largest (Figure 6d, e), further demonstrated that the multi-zonal vOTUs share genetic information with more vOTUs than local and regional vOTUs, suggesting the viral differentiation driven by niche-selection could increase viral speciation. Furthermore, the multi-zonal intra-zones presented a higher popANI value compare with inter-zones (Figure S12) demonstrated that land use shaped differential evolution direction of viral lineage. Furthermore, genes of regional and multi-zonal vOTUs have a significantly lower pN/pS ratio than those of local vOTUs (**Figure 6f**), indicating that soil viruses are likely under negative selection that could cause the extinction of most drifted vOTUs and thus leading to strong niche partition of soil viral communities when land use changes, thus only a small number of multi-zonal vOTUs with high microdiversity were remained and expanded their lineage to adapt the environmental change. '

6. Line 579: “increase the probability of possessing an adaptive genotype as land use changes”. Can you show what these genotypes are? How important are they?

Answer: We realized the sentence could need more evidence to prove, thus we have removed the sentence in this article.

Methods

1. Line 189: you said, “temperate viral genomes in the LVD were identified through package VIBRANT”. You also induced the prophages using Mitomycin C. My question is, did you assign all the iVLPs to be temperate phage? Are all the iVLP temperate phages? How much of the eVLP is a temperate phage? And should you refer this line to Supplementary table 2?

Answer: We did not assign all the iVLPs to be temperate phages due to many lytic phages could also be released during the treatment. The putative lysogenic phages were informatically assigned by the package VIBRANT. The iVLPs and eVLPs were dereplicated and merged into a database LVD, thus we only count the number of temperate phages in LVD through VIBRANT. We have referred this to Supplementary table 2 in L206.

2. Line 190: “The quality of the genome was assessed using package VIBRANT and CheckV v0.7.0”. Should you also refer this to Supplementary table 2?

Answer: As your suggestion, we have referred this to Supplementary table 3 in L207.

3. Line 192-200: Taxonomy assignment. Although vConTACT2 does not intuitively give you a taxonomy assignment list. However, you can find the taxonomy information in “genome_by_genome_overview.csv” by checking “VC Status and VC Subcluster”. So, my question is, can you explain more why not using vConTACT2 taxonomy results instead of using Demovir? And can you please check whether there is an agreement between the taxonomy assigned by “vConTACT2” and “Demovir”? if not how did you resolve the difference?

Answer: Thank you very much for your professional suggestion. We did check the table genome_by_genome_overview.csv from the output of vConTact2, however, all of vOTUs were unassigned. Therefore, all taxonomy assignment through tool vConTACT2 were determined if the vOTUs were clustered with the reference genomes within the

same family or genus using a customized python script. We also compared the taxonomic assignment based on vConTACT2 and Demovir, the Demovir can provide the same results at family level as vConTACT2 when the vConTACT2 can classify the vOTU. We have modified the description in L219-224.

4. Line 209: “Estimation of viral population relative abundances”. What is the virus length cutoff you used for this analysis? Previously (in line 177), you mentioned about “Contigs > 10 kb or circular contigs < 10 kb were obtained”, but I am not sure whether you use this cutoff for all downstream analysis. Therefore, it would be useful to make it clear.

Answer: We use the threshold for all download analysis. We have modified it in L186.

Figures

1. Figure 1A and 1B, I would personally put these two in supplementary. These are informative figures, but not really important to be put in the main manuscript.

Answer: As your suggestion. We have revised the Figure 1A and 1B as Figure S2 and Figure S4 respectively.

2. Figure 1C, it is really hard to see the pie slices <10%. You can either 1) show the top 10 virus families, or 2) separate from “unassigned”, you can combine those <10% as “Others”.

Answer: We have modified it in Figure 1a. We combine those < 10% as ‘Others’

Figure 3B, can you please put the coding/naming in figure caption.

Answer: We have added it in the Figure 3B.

Figure 4, this is a good figure. However, I think it would be helpful if you can show the alpha diversity of lysogenic across three land uses too.

Answer: Thanks. The alpha diversity of lysogenic phages have been supplemented as

Figure 4c.

Figure 5B, why did you show more bacterial-class-links for viruses in comparison to the bacterial composition in figure 5A? I would make two figures consistent and put the full list in the supplementary table.

Answer: Thank you very much for your professional suggestion. We have modified the Figure 5B to keep the consistency between figure 5A and figure 5B.

Figure 5C, do the circle sizes mean anything?

Answer: The size of circle represents Pearson Correlation Coefficient (r) among different environmental factors. We have added the description in the caption of Figure 5C.

Supplementary

Line 9-18: Should you put this explanation to Figure S1 caption?

Answer: Done

Figure S4, I would explain in the figure caption the abbreviation of land use (i.e. FO, AG and UG) and name coding for x-axis.

Answer: We have provided the description of the abbreviation of land use types in Figure S5 and Figure S6.

Figure S5, can you also add the confidence intervals in the PCoA plot?

Answer: Thank you very much for your professional suggestion. The confidence intervals 0.8 were added to the PCoA plot (see below and Figure S8).

Figure S6, I would explain in the figure caption the abbreviation of land use (i.e. FO, AG and UG).

Answer: We have provided the description of the abbreviation of land use types in Figure S7 and Figure S8.

Supplementary table 2, please check the pdf when you convert from excel. There seems to be an issue with the converted table format.

Answer: We provided a new pdf of this table, and we also uploaded the table in excel format, we would like suggest you to read these files with excel format.

Supplementary tables, it is advanceable for the authors to add a footnote on the content of the table. For example, supplementary table 3, what does it mean by Density, Internal weight, External weight, Quality? Please check all supplementary tables

Answer: We have a footnote in Table S4 and we have checked all tables.

Reviewers' Comments:

Reviewer #1:

Remarks to the Author:

The authors have extensively revised their original manuscript and have addressed all of my original comments. I have only one major remark regarding the new prophage analysis included in Figure 4e and described in L439 - L443. My original suggestion was to use the metagenomes (i.e. those generated from non-fractionated samples - the same ones used to generate figure 5a) to identify integrated phages in bacterial sequences to assess the extent at which the VLPs characterized in the viromics approach generated from the induction of lysogenic viruses. The motivation behind this suggested analysis was to confirm if sequences detected in mitomycin-C-treated viromes were more likely to have generated from prophages. Instead, the authors looked for integrated phages in the viromes, which doesn't make much sense considering that the 0.22 um filtration should have depleted the bulk of cellular microorganisms present in the soil samples. As such, I find surprising that the authors were able to find several integrated phages in the assembled sequences from the viromes. Are they integrated phages in nanobacteria (previous paper have shown that 0.22 um viromes have an enrichment of CPR, e.g. <https://journals.asm.org/doi/full/10.1128/mSystems.01205-20>)? Or maybe they are derived from relic DNA that wasn't digested by the DNase treatment? Or could this be a sign of the limitations of the bioinformatic tools used to identify prophages? The authors should comment on the possible origin for these integrated phages in their viromes.

Minor comments:

L894 - there's only one set of asterisks in the figure

Figure 3 - What are "special"? I think it means they were uniquely found in a single zone. Relabel.

L920 - Missing what's the P-value threshold specified by ** - relevant for panels a and b

L487 - define what you mean by local and regional (I am assuming local = found in one single site, and regional = found in multiple sites within a single region).

Figure 6d - y-label should say "Number of links" or "Number of edges". Help the reader quickly understand what your plots are displaying.

Reviewer #2:

Remarks to the Author:

The authors addressed most of my comments. I have a few more as follows.

1. The authors may consider re-arranging the sentences in the abstract to help with the flow. For example, start with the observation of significant viral community structures in soils with different types of land use→that may relate to selection on different soil vOTUs→ what factors that potentially contribute to the selections (pH, disturbance, soil moisture...).
2. Please avoid overstating the research significance. For example, 'mechanistic understandings of ...ecosystem services' in line 46. It is not immediately clear what ecosystem services of soil viruses were studied here. If there are and I missed, please include the supporting results in the abstract and may consider saying it directly. if not, please rewrite the statement.
3. Line 64, expressing AMG is only one of the examples of virus-mediated host metabolism.
4. Line 354, bracket '> 1,500 bp'.
5. Line 354-358, grammar check.
6. A follow-up of my previous comment 30 about comparing iVLPs and eVLPs. Negative result can still be important. If we mention no significant difference was observed when comparing iVLPs and eVLPs of each land use type, it may help to explain why 'they are merged and dereplicated' (line 360).
7. Please note that Family Siphoviridae, Podoviridae and Myoviridae are currently abolished in the latest release of ICTV. Please double check this and update accordingly.

Reviewer #1

The authors have extensively revised their original manuscript and have addressed all of my original comments. I have only one major remark regarding the new prophage analysis included in Figure 4e and described in L439 - L443. My original suggestion was to use the metagenomes (i.e. those generated from non-fractionated samples - the same ones used to generate figure 5a) to identify integrated phages in bacterial sequences to assess the extent at which the VLPs characterized in the viromics approach generated from the induction of lysogenic viruses. The motivation behind this suggested analysis was to confirm if sequences detected in mitomycin-C-treated viromes were more likely to have generated from prophages. Instead, the authors looked for integrated phages in the viromes, which doesn't make much sense considering that the 0.22 um filtration should have depleted the bulk of cellular microorganisms present in the soil samples. As such, I find surprising that the authors were able to find several integrated phages in the assembled sequences from the viromes. Are they integrated phages in nanobacteria (previous paper have shown that 0.22 um viromes have an enrichment of CPR, e.g. <https://journals.asm.org/doi/full/10.1128/mSystems.01205-20>)? Or maybe they are derived from relic DNA that wasn't digested by the DNase treatment? Or could this be a sign of the limitations of the bioinformatic tools used to identify prophages? The authors should comment on the possible origin for these integrated phages in their viromes.

Answer: Thank you very much for your suggestions sorry for our misunderstanding. As suggestion, we investigated the integrated prophages in metagenomes using VIBRANT, and detected only 45 prophages, of which only 22 prophages presented an overlap with the vOTUs detected in the viromes. We think the small data size may not be sufficient to confirm if sequences detected in mitomycin-C-treated viromes were more likely to have generated from prophages. We fully agreed to your speculations, that the presence of prophages in viromes could be integrated prophages in the nanobacteria, or the relic DNA that were carried by bacterial extracellular vesicles and could be not entirely

digested by DNase. In addition, the prophages can also carry host DNA fragment during lateral transduction. We have added these comments in line 315-319.

Reference: John Chen^{1,*},†, Nuria Quiles-Puchalt^{2,*}, Yin Ning Chiang et al. Genome hypermobility by lateral transduction. Science, 12 October 2018, 362(6411):207-212, doi:10.1126/science.aat5867.

Minor comments:

L894 - there's only one set of asterisks in the figure

Answer: We have revised it.

Figure 3 - What are “special”? I think it means they were uniquely found in a single zone. Relabel.

Answer: We have relabel the term as “Unique”, we have explained the term in Material and Methods. We also added the explaining in line 950-954.

L920 - Missing what's the P-value threshold specified by ** - relevant for panels a and b

Answer: We have added it in line 965.

L487 - define what you mean by local and regional (I am assuming local = found in one single site, and regional = found in multiple sites withing a single region).

Answer: Your assumptions are right, and these terms were defined in Material and Methods (section: Classifying multi-zonal, regional, and local vOTUs) and.

Figure 6d - y-label should say “Number of links” or “Number of edges”. Help the reader quickly understand what your plots are displaying.

Answer: We have revised it in Figure 6d.

Reviewer #2

The authors addressed most of my comments. I have a few more as follows.

1. The authors may consider re-arranging the sentences in the abstract to help with the flow. For example, start with the observation of significant viral community structures in soils with different types of land use→that may relate to selection on different soil vOTUs→ what factors that potentially contribute to the selections (pH, disturbance, soil moisture···).

Answer: Thank you very much for your suggestions. We have re-arranged the sentences in abstract.

2. Please avoid overstating the research significance. For example, ‘mechanistic understandings of ...ecosystem services’ in line 46. It is not immediately clear what ecosystem services of soil viruses were studied here. If there are and I missed, please include the supporting results in the abstract and may consider saying it directly. if not, please rewrite the statement.

Answer: Thank you very much for your suggestions. The ecosystem services of soil viruses were not immediately clear via this research. We have rewritten the statement in L44-45.

3. Line 64, expressing AMGs is only one of the examples of virus-mediated host metabolism.

Answer: We have rewritten the sentence in line 64 as “Viruses are extremely abundant and diverse biological entities on earth, playing vital roles in affecting soil microbiota and functions^{3,4} via regulating microbial community dynamics⁵, reprogramming host metabolism during infection^{6,7}, and serving as vectors of horizontal gene transfer³.”

4. Line 354, bracket ‘> 1,500 bp’.

Answer: We have rewritten the sentence in line 117.

5. Line 354-358, grammar check.

Answer: We have revised these sentences in line 117-121.

6. A follow-up of my previous comment 30 about comparing iVLPs and eVLPs. Negative result can still be important. If we mention no significant difference was observed when comparing iVLPs and eVLPs of each land use type, it may help to explain why 'they are merged and dereplicated' (line 360).

Answer: Thank you very much for your suggestions. We have added the result in Figure S5 and Line 155-157.

7. Please note that Family Siphoviridae, Podoviridae and Myoviridae are currently abolished in the latest release of ICTV. Please double check this and update accordingly.

Answer: Thank you very much for your suggestions. We also noticed that the latest release of ICTV (2021) modified a substantial of taxonomic names compared to ICTV 2020, some genus from ICTV 2020 were proposed as family, such as genus *Peduovirinae* in family *Myoviridae* (ICTV 2020) were reclassified as family *Peduoviridae* (ICTV 2021), and most of the genera in the family *Siphoviridae* (ICTV 2020) have not yet been assigned to a family in ICTV 2021. Since 1971, the viral taxonomic framework was constantly modified every year, suggesting that the viral taxonomy is highly variable. Furtherly, although the family *Siphoviridae*, *Podoviridae* and *Myoviridae* have been abolished in ICTV 2021, the vConTACT2 and demovir2 have not yet updated their referenced taxonomic list based on ICTV 2021, it would be very complex to update the results since all of viral taxonomy were predicted based on the bioinformatic tool vConTACT2 and demovir2 in this study. Overall, we choose to explain the inconsistent between our predictions and ICTV 2021 in Materials and Methods (see line 561-564). We hope the vConTACT2 and demovir updated their reference taxonomic framework of viruses in the future based on the latest release of ICTV, and we will reanalysis the soil taxonomic compositions.